# Role Diversity Matters: A Study of Cooperative Training Strategies for Multi-Agent RL

## Abstract

Cooperative multi-agent reinforcement learning (MARL) is making rapid progress for solving tasks in a grid world and real-world scenarios, in which agents are given different attributes and goals. For example, in Starcraft II battle tasks, agents are initialized with the various move, defense, and attack abilities according to their unit types. Current researchers tend to treat different agents equally and expect them to form a joint policy automatically. However, ignoring the differences between agents in these scenarios may bring policy degradation. Accordingly, in this study, we quantify the agent's difference and study the relationship between the agent's role and the model performance via **Role Diversity**, a metric that can describe MARL tasks. We define role diversity from three perspectives: policy-based, trajectory-based, and contribution-based to fully describe the agents' differences. Through theoretical analysis, we find that the error bound in MARL can be decomposed into three parts that have a strong relation to the role diversity. The decomposed factors can significantly impact policy optimization on parameter sharing, communication mechanism, and credit assignment strategy. Role diversity can therefore serve as a flag for selecting a suitable training strategy and helping to avoid possible bottlenecks on current tasks. The main experimental platforms are based on **Multiagent Particle Environment (MPE)** and **The StarCraft Multi-Agent Challenge (SMAC)**, with extensions to ensure the requirement of this study are met. Our experimental results clearly show that role diversity can serve as a robust description for the characteristics of a multi-agent cooperation task and help explain the question of why the performance of different MARL training strategies is unstable according to this description. In addition, role diversity can help to find a better training strategy and increase performance in cooperative MARL.

## 1 Introduction

Recently, multi-agent reinforcement learning (MARL) has captured people's attention due to its impressive achievements in the field of super human-level intelligence in video games [3, 6, 52, 57], card games [7, 25, 44, 59], and real-world applications [62–64]. These achievements have benefited substantially from the success of single-agent reinforcement learning (RL) [14, 15, 30, 42, 43] and rapid progress of MARL from both the competitive and the cooperative side.

On the competitive side, existing works focus on game theory among adversary agents with guaranteed policy convergence via theoretical analysis [5, 17, 27, 29, 56]. Whereas the achievements on the cooperative MARL are more based on empirical results in cooperative multi-agent system (MAS) [4, 21, 28, 39, 46, 51, 61]. One key problem of cooperative MARL is that whether one algorithm is better than another depends on the MARL tasks as showed in Fig. 1a. Current researches focus on developing algorithms on the tasks they are good at but lack the study of why the performance declines on other tasks [16, 35, 55, 55, 58]. Even adopting the state-of-the-art algorithms does not guarantee a strong performance [13, 35, 48, 55, 58]. This may due to the varying characteristic (e.g agent's attributes and goals) of MARL tasks and scenarios, one single algorithm is not able to cover them all, which means we have to change the training strategy according to the current scenario.

From this perspective, we need to find a metric to describe different MARL tasks and use this description to help determine the best strategy combination as showed in Fig. 1b. Considering that

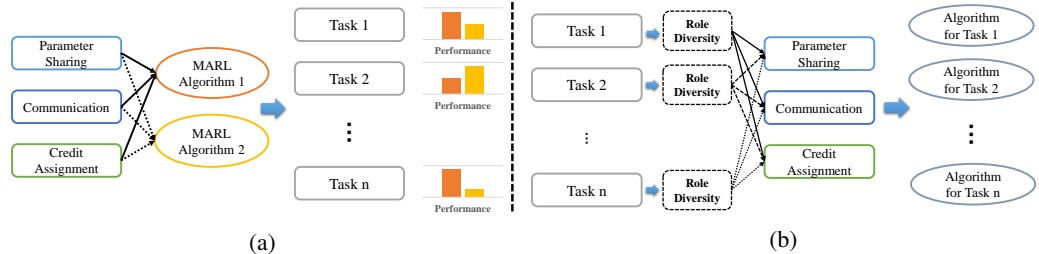

Figure 1: (a) Performance of MARL algorithm 1 & 2 combined with different parameter sharing, communication, and credit assignment strategies varies from candidates 1 to $n$. (b) Using Role Diversity to describe each task, we can ensure the best combination of different strategies.

the main component of MAS is the agents, we propose a new definition **Role** in MAS to quantify the agents' difference and use **Role Diversity** to describe MAS. We then analyze how the role diversity impacts the MARL both theoretical and experimental. For theoretical analysis, we use the decomposed estimation error of the joint action-value function $Q_{tot}$ in MARL to discuss how role diversity impacts the policy optimization process. The experiment further verifies the theoretical analysis that the role diversity is strongly related to the model performance and can serve as a good description of a MAS. As shown in Fig. 1, with the role diversity description of each task, we can now avoid possible bottleneck of a MARL algorithm with the combination of different parameter sharing, communication, and credit assignment strategies. With the definition of role diversity and the analysis of its impact on MARL, we can also explain the question of why the model performance varies across different tasks.

Role diversity are defined from three aspects: policy-based, trajectory-based, and contribution-based in Sec. 3 which are measured by action distribution, observation overlap, and Q/state value diversity. Through theoretical analysis, we find each type of role diversity has different impact to different terms of decomposed estimation error: algorithmic error, approximation error, and statistical error (Sec. 4). We conduct comprehensive experiments covering three main topics in MARL: parameter sharing , communication , and credit assignment in Sec. 5 and provide a set of guideline on choosing MARL training strategy in Sec. 6. The main experiments are conducted on MPE [28] and SMAC [39] benchmarks, covering a variety of scenarios with different role diversity. The impact of role diversity is evaluated on representative MARL algorithms including IQL [49], IA2C [31], VDN [48], QMIX [35], MADDPG [28], and MAPPO [58], covering independent learning methods, centralised policy gradient methods, and value decomposition methods. The experiment results prove that the model performance of different algorithms and training strategies is largely dependent on the role diversity. Scenarios with large policy-based role diversity prefer no parameter sharing strategy. Communication is not needed in scenarios with large trajectory-based role diversity. Learnable credit assignment modules should be avoided when training on scenarios with large contribution-based role diversity.

The key contributions of this study are as follows: First, the concepts of the role and role diversity are defined to describe MARL tasks. Second, a theoretical analysis of how role diversity impacts MARL policy optimization with estimation error decomposition is built. Third, role diversity is proven to be strongly related to performance variance when choosing different training strategies, including parameter sharing, communication, and credit assignment on the MARL benchmarks. Finally, a set of guidelines for selecting a training strategy based on role diversity is provided.

## 2 RELATED WORK

Researches on the development of cooperative MARL algorithms are mainly in three aspects: parameter sharing, communication and credit assignment.

For parameter sharing, the common approach in cooperative MARL is to fully share the model parameters among the agents [16, 35, 48, 55]. In this way, the policy optimization can benefit from the shared experience buffer with samples from all different agents, providing a higher data efficiency. However, it has also been noted in recent works that parameter sharing is not always a good choice [8, 34, 50]. In some scenarios, a selective parameter sharing strategy, or even no parameter sharing,

can significantly benefit agent performance and surpass the full parameter sharing. However, the question of why different parameter sharing strategies have different impacts on different scenarios remains open. In this study, we find that the role diversity can serve as a strong signal for selecting the parameter sharing strategy.

The communication mechanism is an intrinsic part of the multi-agent system (MAS) framework [18, 19, 23, 28, 47]. It provides the current agent with essential information of other agents to form a better joint policy, which substantially impacts the final performance. In some cases, communication restrictions exist, which hinder us from freely choosing communication methods [28, 39]; in most cases, however, the communication is available and it is optional on when to communicate and how to ingest the shared information [19, 45]. We present a comprehensive study on the relationship between role diversity and information sharing via communication mechanisms and demonstrate that role diversity determines the necessity of communication.

For the credit assignment, most cooperative MARL algorithms adopt Q-learning or policy gradient as the basic policy optimization method, which is combined with an extra value decomposition module [16, 35, 48, 55] or shared critic function [13, 28, 58] to optimize the individual policy. Some other works find that leveraging the reward signal is unnecessary; however, optimizing the individual policy independently (independent learning, IL) can still get a strong joint policy [34, 49]. It then becomes slightly difficult to decide which credit assignment method (including IL) is better as there is no single method in cooperative MARL that is robust and always outperforms others (compared to PPO [43] or SAC [14] in single-agent RL) on different tasks. In this study, we contend that role diversity is the key factor that impacts the performance of different credit assignment strategies.

In the next section, we present the role definition from three aspects including policy-based, trajectory-based, and contribution-based, and propose the measurement of different role diversity types to describe a MARL task.

## 3 ROLE DIVERSITY

Using role to describe the characteristic of the agents in the MARL context has been proven to be effective in many recent works [8, 24, 54, 55]. However, the definition of the role concept remains largely unclear. In work [55], the role is defined as the higher-level option in the hierarchical RL framework [20]. In work [8], the role is defined as the environmental impact similarity of a random policy. These definitions are intuitive and cannot accurately describe the role difference. In this study, we attempt to define the role in a more comprehensively way from three different aspects: policy-based, trajectory-based, and contribution-based. More specific scenario-based illustration can be found in Fig. 7 and Fig. 8. With our refined role, a strong relationship between role diversity and the MARL optimization process can be built and the performance variance can be further explained.

### 3.1 POLICY-BASED ROLE

In MARL, different agents output different actions based on its current status. As common sense would indicate, actions taken at the same timestep can indicate different roles [55]. However, there are many exceptions. For instance, if we have two soccer players passing a ball to each other repeatedly [21], although the action is different at each time step, the roles of these two soccer players can be very similar from the perspective of the whole soccer game. Therefore, it is not sufficient to distinguish the role difference based on a single timestep. Instead, we contend that this difference should be defined based on a period. As this role is purely based on policy distribution $\pi$, we refer to it as a policy-based role.

Specifically, we define the policy-based role as the statistics of the actions' frequency over a period, which is $n$ steps backward and forward from the current timestep. Here, $n$ is the time interval that is half the length of the total time. More details can be found in Fig. 8a, where we provide a real scenario from SMAC. Policy-based role difference can be represented as follows:

$$r_T^a = \frac{1}{2n+1} \sum_{t=T-n}^{T+n} \pi_t^a \tag{1}$$

where $T$ represents the current timestep, $n$ is the time interval, $a$ is the agent index, $\pi$ is the policy distribution. We adopt symmetrical KL divergence to measure the distance of different policy-based

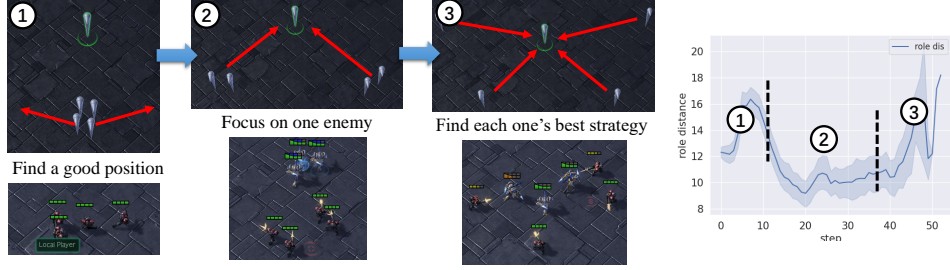

(a) Policy-based role in one episode on 4m_vs_3z

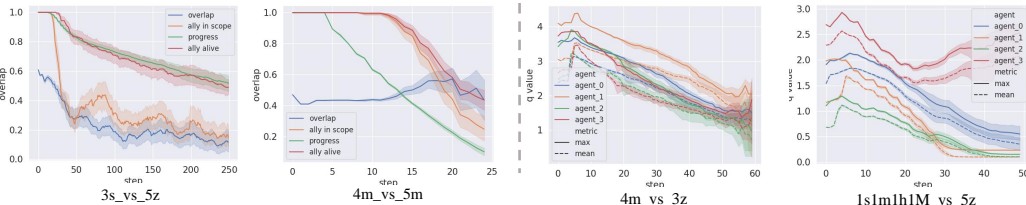

(b) Observation overlap percentage curve

(c) Q value of different agents curve

Figure 2: (a) An illustration of how policy-based role difference varies in one game(4m_vs_3z). A detailed explanation of how it varies can be found in Sec. 3.1. (b) Instance curve of the trajectory-based role difference in different battle scenarios according to observation overlap. Trajectory-based role difference is larger in 3s_vs_5z but smaller in 4m_vs_5m. (c) Contribution-based role diversity in different battle scenarios represented by Q value. The contribution-based role diversity is larger in 1s1m1h1M_vs_5z but smaller in 4m_vs_3z.

roles. The total role distance of $A$ agents can be computed as follows:

$$d_T^p = \frac{1}{\sum_0^{A-1}} \sum_{a_0=0}^{A} \sum_{a_1=a_0}^{A} \left(KL(r_T^{a_0}|r_T^{a_1}) + KL(r_T^{a_1}|r_T^{a_0})\right) \qquad (2)$$

where $d_T^p$ represents the policy-based role distance at timestep $T$, $A$ represents the total agent number, and *KL* represents Kullback–Leibler divergence.

We also provide a case study of how policy-based role diversity varies in Fig. 2a. From a real battle scenario (4m_vs_3z) taken from SMAC, we find three stages including ***Find a good position***, ***Focus on enemy*** and ***Find each one's best strategy***. In stage 1, agents try to find their own best location; the role diversity is large. In stage 2, agents focus on the same enemy target; the policies become similar and the role diversity is decreased. In stage 3, the formation of the agents is broken up by the enemies. Policy-based role diversity again increases as each agent is required to find its own best strategy to deal with its current situation.

## 3.2 TRAJECTORY-BASED ROLE

A policy-based role only considers the role diversity from the action distribution perspective. A slight action difference may not diversify policy-based role; however, this difference can be enlarged by time, which eventually results in a de facto behavior difference. The most significant phenomenon caused by this is the agent's trajectory. Again, consider a soccer game where players repeatedly pass a ball to each other; these players may have a similar policy-based role, but their trajectories are different. This difference is enhanced by the partial observation setting that exists in many popular cooperative MARL environments [39, 46, 61] as the partial observed input from different agents shares a less common pattern when the vision scope is smaller. Therefore, trajectory-based role diversity is an important supplement to policy-based role diversity.

Generally speaking, we can define the trajectory-based role as the record of the agent's movement. However, the determination of the extent to which two trajectories differ is not a straightforward matter. To measure this difference, we use an indirect metric called observation overlap percentage.

Using observation overlap to measure the trajectory-based role difference is: *I.* easy to compute, *II.* able to utilize a constant scale from 0 to 1 and *III.* strongly related to observation scope, which means that two trajectories can have varying role distances. For instance, in an $A$-agent multi-agent system, the trajectory difference can be computed as follows:

$$d_T^{traj} = \frac{1}{\sum_0^{A-1}} \sum_{a_0=0}^{A} \sum_{a_1=a_0}^{A} d_T^{a_0,a_1}. \tag{3}$$

Here, $d_T^{a_0,a_1}$ is the observation overlap percentage of the total area between agent $a_0$ and agent $a_1$ at timestep $T$. An example of calculating the observation overlap percentage in SMAC [39] can be found in Appx. C. We provide the observation overlap percentage curves in Fig. 2b to demonstrate how trajectory-based role diversity varies in different scenarios.

### 3.3 Contribution-Based Role

To help ensure the authenticity of the modern MAS, agents are initialized with different attributes in recently proposed MARL environments [21, 28, 39, 46]. For instance, in [21], the roles of forward and goalkeeper are quite different and characterized by different observation scopes, action spaces, and reward functions. The type differences are easy to notice but hard to define, as is the role distance between them.

Here, we use an indirect variable called contribution to measuring the type difference. In cooperative MARL, the final target is to obtain an optimal joint policy consisting of $A$ individual policies. To do this, the credit assignment strategy is proposed in [13, 35, 48] to leverage the reward signals to each agent to achieve individual policy optimization. A good credit assignment strategy should be able to leverage the reward signal in a manner equal to each agent's contribution to the global reward. In this way, the Q value (Q function in off-policy RL) or state value (critic function in on-policy RL) of each agent can be estimated based on the leveraged reward signal. From this perspective, the Q value or the state value can be regarded as the agent's contribution to the team. Generally speaking, we use the Q value or state value to measure the contribution of a single agent. Contribution-based role diversity can be computed as follows:

$$d_T^{cont} = \frac{1}{\sum_0^{A-1}} \sum_{a_0=0}^{A} \sum_{a_1=a_0}^{A} \frac{d_T^{v_{a_0},v_{a_1}}}{max(v_{a_0}, v_{a_1})}. \tag{4}$$

Here, $v$ is the Q value or state value of the policy output and $d_T^{v_{a_0},v_{a_1}}$ is the absolute value difference between the agents' output. In addition, we use max function to keep the range of contribution-based role diversity from 0 to 1. We provide the Q value (mean & max) curves in Fig. 2c to demonstrate that the contribution-based role diversity can vary a lot in different scenarios.

More detailed role definition, connection, and application discussion can be found in Appx. C, D and E. In this paper, all the role diversity values and curves including $d_T^p$, $d_T^{traj}$, and $d_T^{cont}$ come from VDN[28] and separated training with no communication for robust performance and training-efficient.

## 4 Theoretical Analysis

In this section, we use a simple scenario as an example to illustrate the role diversity. Suppose each agent makes individual observations and the learning procedure of all agents is independent. We provide finite-sample analysis for the estimation error of the joint action-value function and identify the terms corresponding to the role diversity. We denote $Q_{tot}^*$ and $Q_i^*$ as the optimal joint and individual Q-function respectively and write $\|\cdot\|_{p,\mu}$ as the $L_p$ norm with respect to a probability measure $\mu$. Motivated by [48] and [35], we consider a simple case:

$$Q_{tot}^* \approx F(\mathbf{Q}^*) = \mathbf{w}^\top \mathbf{Q}^* \quad \text{with} \quad \mathbf{w} \in \Delta_n, \ \mathbf{Q}^* = (Q_1^*, \dots, Q_n^*),$$

where $n$ is the number of agents and $\Delta_n$ is the $(n-1)$-dimensional probability simplex. Here, the credit assignment function $F$ is a weighted sum of $Q_i^*$, $i \in [n]$ with non-negative weights. We then study the excess risk that is the error gap between the estimated and optimal solution:

$$Err \quad = \quad \|Q_{tot}^* - \hat{\mathbf{w}}^\top \mathbf{Q}_t\|_{1,\mu} - \|Q_{tot}^* - (\mathbf{w}^*)^\top \mathbf{Q}^*\|_{1,\mu},$$

where $\mathbf{w}^*$ and $\hat{\mathbf{w}}$ are the optimal and estimated weights and $\mathbf{Q}_t$ is the output of FQI algorithm at the iteration $t$. We further denote $\mathcal{Q}$ as the space of individual Q-functions and write $\omega(\mathcal{Q}) = \sup_{Q \in \mathcal{Q}} \inf_{Q' \in \mathcal{Q}} \|Q' - TQ\|_{2,\nu}^2$. In Sec. B.3, we prove that

$$
\begin{aligned}
Err \quad \leq \quad & \sqrt{n} \times \|\mathbf{w}^* - \hat{\mathbf{w}}\| \times \left\| \sqrt{\mathrm{Var}_n(\mathbf{Q}^*)} \right\|_{1,\mu} \\
& + \frac{4\phi_{\mu,\nu}\gamma}{(1-\gamma)^2}\sqrt{\omega(\mathcal{Q})} + O(\sqrt{\frac{\ln N_0}{N}}) + \frac{4\gamma^{T+1}}{(1-\gamma)^2}M,
\end{aligned}
\tag{5}
$$

where $N$ is the sample size and $T$ is the number of iterations. In addition, $\phi_{\mu,\nu}$ is the concentration coefficient and $N_0$ represents the $1/N$-covering number of $\mathcal{Q}$. Please refer to Sec. B.1 for the detailed definitions.

The first term on the RHS of (5) reflects the benefit of credit assignment that is strong related to the **Contribution-Based Role** (Sec. 3.3). When $\mathrm{Var}_n(\mathbf{Q}^*)$ is non-negligible, minimizing $\|\mathbf{w}^* - \hat{\mathbf{w}}\|$ can significantly decrease the excess risk. The second term that involves $\omega(\mathcal{Q})$ stands for the approximation error caused by functional approximation in $\mathcal{Q}$. It depends on the concentration of the sample and the scale of the hypothetical space. The remaining two tems are statistical error and algorithmic error. If the sample size is sufficiently large and the learning time is long enough, they can be arbitrarily small. In Sec. B.5, we assume $Q$ is a sparse ReLU network and $TQ$ is a composition of Hölder smooth functions, and analyze the convergence rate as $N, T \to \infty$.

Next, we demonstrate that the variance term $\mathrm{Var}_n(\mathbf{Q}^*)$ is related to both the **Policy-Based Role** (Sec. 3.1) and the **Trajectory-Based Role** (Sec. 3.2). We consider the case in which all agents share one estimated Q-function and denote the optimal share Q-function as $\bar{Q}^*$. Sec. B.4 proves that

$$
\begin{aligned}
Err_{share} \quad \leq \quad & \left\| \sum_{i=1}^{n} \mathbf{w}_i^*(Q_i^* - \bar{Q}^*) \right\|_{1,\mu} + \sqrt{n} \times \|\mathbf{w}^* - \hat{\mathbf{w}}\| \times \left\| \sqrt{\mathrm{Var}_n(\bar{\mathbf{Q}}^*(\mathbf{z},\mathbf{u}))} \right\|_{1,\mu} \\
& + \frac{4\phi_{\mu,\nu}\gamma}{(1-\gamma)^2}\sqrt{\omega(\mathcal{Q})} + O(\sqrt{\frac{\ln N_0'}{nN}}) + \frac{4\gamma^{T+1}}{(1-\gamma)^2}M,
\end{aligned}
\tag{6}
$$

where $N_0'$ is the $1/(nN)$-covering number of $\mathcal{Q}$ and

$$
\bar{\mathbf{Q}}^*(\mathbf{z}, \mathbf{u}) = \big( \bar{Q}^*(\mathbf{z}_1, \mathbf{u}_1), \bar{Q}^*(\mathbf{z}_2, \mathbf{u}_2), \dots, \bar{Q}^*(\mathbf{z}_n, \mathbf{u}_n) \big).
$$

The first term on the RHS of (6) stands for the bias caused by parameter sharing. If all $Q_i^*$ are the same, the bias will disappear. Therefore, the **Policy-Based Role** is related to this bias. Second, the variance $\mathrm{Var}_n(\bar{\mathbf{Q}}^*)$ here is caused by the trajectory diversity. To reduce this term, we should should ensure that all agents have similar observations. In addition, the **Trajectory-Based Role** measures the concentration of all agents' support set. It is therefore natural to group the highly overlapped agents into one sub-joint agent via communication mechanism. This can be compared to the separate case in (5), where approximation error and the learning error are the same. In Sec. B.5, we show that the parameter sharing improves the convergence rate of the statistical error via sample pooling, while the communication decreases the convergence rate by activating more input variables.

## 5 EXPERIMENT

In this section, we mainly demonstrate how model performance varies with role diversity and how to adjust the training strategy in the context of cooperative MARL. The experimental results show the following: 1. that the performance of different parameter sharing strategies is strongly related to the **Policy-Based Role** (Sec. 5.1). 2. that the benefit brought by different communication mechanisms can be easily affected by the **Trajectory-Based Role** (Sec. 5.2). 3. that the performance of the credit assignment method, or the centralized training strategy, is largely dependent on the **Contribution-Based Role** (Sec. 5.3). 4. that the choosing of training strategies should be determined by the scale of role-diversity for different scenarios. The main experimental platforms are MPE [28] and SMAC [39]. Extensions are made to fulfill the requirements of parameter sharing and the communication mechanism, these include separated training of policy in Sec. 5.1 and information exchange among agents in Sec. 5.2. All results come from eight random seeds. More details regarding the experimental settings can be found in the appendix.

| Benchmark | Scenario | Role Diversity | Warm-up | No shared | Partly shared | Shared |
|---|---|---|---|---|---|---|
| MPE | SimpleSpread | 14.1 | -598.3 | +137.0 / +142.9 | +149.0 / +176.4 | **+154.1 / +198.0** |
| | Tag | 17.8 | 3.8 | +43.4 / +57.3 | **+47.0 / +60.9** | +48.8 / +59.2 |
| | Adversary | 18.3 | 10.7 | +5.2 / +5.7 | **+6.2 / +6.6** | +5.4 / +5.9 |
| | DoubleSpread-2 | 17.6 | 7.3 | **+47.8 / +53.2** | +28.6 / +34.6 | +3.6 / +15.9 |
| | DoubleSpread-4 | 19.5 | 22.0 | **+29.5 / +192.4** | +12.0 / +91.3 | +11.4 / +5.3 |
| SMAC | 4m_vs_5m | 1.5 / 9.1 | 6.5 | +3.6 / +4.4 | +4.3 / +5.0 | **+5.4 / +6.1** |
| | 3s_vs_5z | 2.7 / 18.7 | 5.4 | +7.5 / +11.0 | +6.6 / +9.6 | **+8.2 / +11.8** |
| | 2m | 3.1 / 12.2 | 6.0 | +9.2 / +11.1 | +15.5 / +15.6 | **+18.1 / +17.6** |
| | 4m_vs_4z | 3.3 / 19.3 | 4.4 | + 8.8 / +12.7 | **+ 10.5 / +14.7** | +5.4 / +8.4 |
| | 4m_vs_3z | 3.8 / 12.1 | 7.2 | +12.4 / +12.1 | **+12.5 / +12.5** | +11.9 / +12.3 |
| | 3s_vs_4z | 5.2 / 32.5 | 4.8 | **+2.2 / +4.5** | +1.7 / +2.7 | +0.9 / +1.2 |
| | 1c1s1z_vs_1c1s3z | 8.7 / 22.0 | 11.8 | **+4.1 / +6.1** | +3.7 / +5.9 | +2.7 / +5.4 |
| | 1s1m1h1M_vs_3z | 2.4 / 13.2 | 16.2 | +3.4 / + 3.4 | +3.6 / +3.6 | **+3.4 / +3.6** |
| | 1s1m1h1M_vs_4z | 2.7 / 15.8 | 8.2 | **+7.8 / +11.6** | +6.5 / +10.9 | +5.3 / +10.0 |
| | 1s1m1h1M_vs_5z | 6.2 / 22.5 | 6.2 | **+6.4 / +9.1** | +4.2 / +8.5 | +3.7 / +6.1 |

Table 1: Performance of three parameter sharing strategies on different scenarios. *Warm-up* refers to the reward value point where the strategies start to differentiate. + represents the additional reward gained based on warm-up performance. The left side and right side of the */* represent the reward gained at the half training steps and the full training steps respectively. The best performance in each scenario is marked in bold red. More detailed analysis can be found in Sec. 5.1.

## 5.1 PARAMETER SHARING

Policy-based role influences the convergence speed and final performance of different parameter sharing strategies in cooperative MARL. The scenarios we choose from the MPE and SMAC benchmarks are simple but diverse, covering policy-based role diversity that ranges from small to large. In table. 1 and Fig. 3, we provide policy-based role diversity and the model performance curve on these chosen scenarios. For the SMAC benchmark, we adopt two metrics to count $r_T^u$ in Eq. 1. The first metric is **real policy diversity**, which treats each action as an independent. The second way is **semantic policy diversity**, which distributes the actions to different groups according to their semantic type (e.g. move & attack). There is no semantic policy-based role in scenarios chosen from the MPE benchmark as all the actions are of the same semantic type (move). The policy-based role diversity is then calculated according to Eq. 2. The base MARL credit assignment strategy we choose in table. 1 is **VDN[48]**, combined with a fully/partly/no parameter sharing strategy. For details of how the partial parameter sharing strategy works, please refer to the Appx. F. We also evaluate other popular credit assignment strategies including **IQL[49], IA2C, MADDPG[28] MAPPO[58], MAA2C** and **QMIX[35]** combined with a fully/no parameter sharing strategy in Fig. 3. **IA2C** and **MAA2C** are the extension of **A2C[31]** on multi-agent scenarios.

Table. 1 outlines the performance of three parameter training strategies including *No shared*, *Partly shared* and *Shared* with the base credit assignment method VDN[48]. Detailed framework and settings can be found in Appx. F. As the policy-based role diversity increases, the performance of the *Shared* strategy is degraded in terms of both convergence speed (half training steps) and the final reward (full training steps). One interesting phenomenon also emerges: the same agent type (e.g. 3s_vs_5z, 4m_vs_4z) does not always indicate small policy-based role diversity, and vice versa (e.g. 1s1m1h1M_vs_3z), which means it is hard to define the role before identifying an adequate policy. Fig. 3 shows the model performances of two parameter sharing strategies including *No shared* and *Shared* with different credit assignment methods. For policy gradient-based methods, we extend the training steps from the standard 2M to 20M (10 times) as the convergence speed of policy gradient-based methods (e.g. MAPPO, MAA2C) is slower than Q value-based methods. From Fig. 3, we find that different credit assignment methods have a slight impact on parameter sharing strategies but the trend in which no parameter sharing strategy achieves performance improvement continues to be present as the policy-based role diversity increases.

Here we conclude that, scenarios with large policy-based role diversity prefer no parameter sharing strategy, and vice versa. Suitable parameter sharing strategy helps obtain faster convergence speed and higher final performance.

## 5.2 COMMUNICATION

More information is better. This principle is common sense in many areas including computer vision or natural language processing. With a bigger dataset and more detailed and accurate annotations, the model can be better optimized. However, in reinforcement learning, the data is sampled using the current policy; in most cases, this policy learning starts with a randomly initialized neural network. In this way, the provision of more information may introduce a burden for the policy optimization and can moreover further degrade the sampled data quality, which gives rise to a vicious

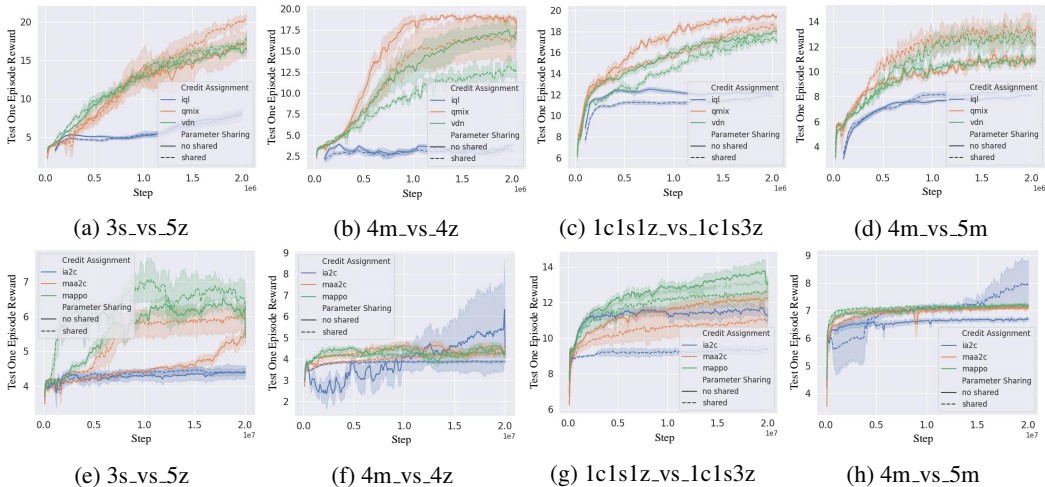

(a) 3s_vs_5z     (b) 4m_vs_4z     (c) 1c1s1z_vs_1c1s3z     (d) 4m_vs_5m

(e) 3s_vs_5z     (f) 4m_vs_4z     (g) 1c1s1z_vs_1c1s3z     (h) 4m_vs_5m

Figure 3: Performance curves include Q value-based(first row) and policy gradient-based(second row) credit assignment with *Shared* and *No shared* parameter sharing strategies.

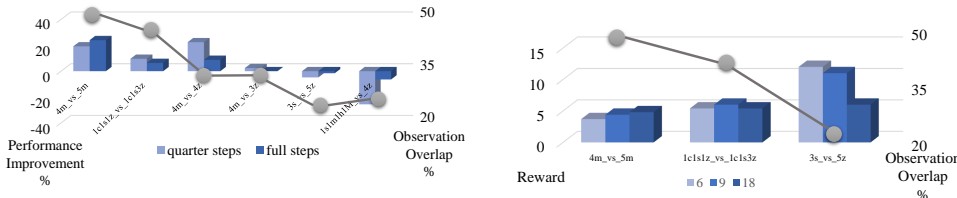

(a) Model performance with or without communication

(b) Model performance with various vision scope

Figure 4: Histogram of (a) model performance when adopting communication mechanism compared to baseline (w/o communication). (b) model performance with different vision scope (6-9-18), where scope 9 is the standard setting in SMAC. Grey dots represent the observation overlap. Larger the overlap, the smaller the trajectory-based role diversity. Detailed analysis can be found in Sec. 5.2

circle. Therefore, it is critical to determine when and how to accept the extra information provided via communication mechanism in the context of cooperative MARL. As discussed in Sec. 4, the pattern of different agents' support sets for policy optimization can determine whether or not the extra information is needed. Notably, the similarity of these support sets is largely dependent on the trajectories of different agents, corresponding to the trajectory-based role diversity defined in Sec. 3.2. Small trajectory-based role diversity corresponds to a similar support set pattern, which means that forming a concentrated input is preferred for policy optimization. Experiment results in table. 7 and Fig. 4 further prove that scenarios with larger observation overlap are more suitable for communication. Detailed setting of communication mechanism on SMAC can be found in Appx. G

To prove that small trajectory-based role diversity prefers obtaining extra information via communication and vice versa, we conduct extra experiments to study the relationship between the pattern of the input observation (support set) and the model performance by shrinking the vision scope ($r$ in eq. 14). The results can be found in table 2. The results show that the model performance is strongly related to the vision scope and the trajectory-based role diversity determines whether the small or large vision is preferred. Small trajectory-based role diversity prefers large vision scope, indicating the similar pattern of support set is better. Large trajectory-based role diversity prefers small vision scope, which means enlarging the pattern difference benefits the policy optimization. This further prove that extra information provided by communication forms a similar pattern of support set which is preferred in scenarios with small trajectory-based role diversity.

### 5.3 CREDIT ASSIGNMENT

The performance of different credit assignment methods is strongly related to the contribution-based role. We mainly focus on three representative Q value-based MARL algorithm: VDN[48], QMIX[35] and IQL[49], and compare their performance on different scenarios that have different contribution-based role diversity measured by Q value according to Eq. 4. The result can be found

| scenario | obs overlap | scope | performance | scenario | obs overlap | scope | performance |
|---|---|---|---|---|---|---|---|
| 1s1m1h1M_vs_3z | 0.41 | 6 | 15.6 / 19.5 / 19.5 | 4m_vs_5m | 0.47 | 6 | 6.3 / 9.2 / 10.4 |
| | | 9 | 16.4 / 19.5 / 19.7 | | | 9 | 6.5 / 10.1 / 10.9 |
| | | 18 ✓ | 16.1 / 19.6 / 19.9 | | | 18 ✓ | 6.8 / 10.9 / 11.1 |
| 1s1m1h1M_vs_4z | 0.25 | 6 | 8.4 / 15.3 / 18.8 | 1c1s1z_vs_1c1s3z | 0.40 | 6 | 11.5 / 15.1 / 17.6 |
| | | 9 ✓ | 8.4 / 15.7 / 19.7 | | | 9 ✓ | 12.3 / 16.0 / 17.8 |
| | | 18 | 7.8 / 11.8 / 15.9 | | | 18 | 12.4 / 15.3 / 17.6 |
| 1s1m1h1M_vs_5z | 0.18 | 6 ✓ | 6.6 / 14.2 / 17.7 | 3s_vs_5z | 0.21 | 6 ✓ | 6.0 / 15.1 / 17.5 |
| | | 9 | 6.3 / 12.6 / 15.3 | | | 9 | 5.4 / 12.9 / 16.4 |
| | | 18 | 5.9 / 8.9 / 10.4 | | | 18 | 5.2 / 9.0 / 12.1 |

Table 2: Different vision scopes (6-9-18) impact the model performance. The scope should be larger than 6, which is the attack scope for agents. ✓ represents the scope with the best performance. Detailed analysis can be found in Sec. 5.2.

| scenario | Q diversity | no shared | | | shared | | |
|---|---|---|---|---|---|---|---|
| | | vdn | qmix | iql | vdn | qmix | iql |
| 1c1s1z_vs_1c1s3z | | 12.3 / 15.9 / 17.9 | 12.9 / 17.8 / 19.4 ✓ | 10.8 / 12.3 / 12.2 | 11.2 / 14.5 / 17.2 | 12.5 / 15.8 / 18.4 ✓ | 9.8 / 11.2 / 11.9 |
| 3s_vs_5z | <0.1 | 5.4 / 12.9 / 16.4 | 4.6 / 13.5 / 17.0 ✓ | 4.6 / 5.1 / 7.9 | 6.0 / 13.6 / 17.2 | 4.2 / 12.9 / 20.0 ✓ | 4.3 / 5.3 / 7.8 |
| 4m_vs_4z | | 4.3 / 13.2 / 17.1 | 4.3 / 18.3 / 18.8 ✓ | 3.3 / 3.2 / 3.7 | 4.6 / 9.8 / 12.8 | 4.3 / 14.8 / 16.5 ✓ | 2.6 / 3.2 / 3.2 |
| 4m_vs_5m | 0.1-0.5 | 6.5 / 10.1 / 10.9 * | 7.0 / 9.9 / 10.9 * | 4.8 / 7.6 / 8.1 | 6.8 / 11.9 / 12.6 * | 6.9 / 12.4 / 13.3 * | 5.1 / 8.1 / 8.5 |
| 4m_vs_3z | | 7.5 / 19.6 / 19.3 * | 6.5 / 19.7 / 19.3 * | 4.5 / 5.7 / 11.1 | 6.3 / 19.1 / 19.5 * | 6.1 / 19.7 / 19.7 * | 4.2 / 4.5 / 5.7 |
| 1s1m1h1M_vs_3z | | 16.4 / 19.6 / 19.6 ✓ | 6.5 / 7.5 / 7.8 | 11.1 / 16.9 / 19.2 | 16.1 / 19.6 / 19.8 ✓ | 9.9 / 9.8 / 8.9 | 12.2 / 17.9 / 19.6 |
| 1s1m1h1M_vs_4z | >0.5 | 8.4 / 16.0 / 19.8 ✓ | 4.9 / 5.1 / 6.1 | 7.4 / 9.0 / 10.7 | 8.1 / 13.5 / 18.2 ✓ | 5.5 / 5.0 / 5.0 | 7.1 / 8.5 / 8.5 |
| 1s1m1h1M_vs_5z | | 6.3 / 12.6 / 15.3 ✓ | 4.2 / 4.2 / 3.6 | 5.5 / 6.1 / 6.5 | 6.2 / 9.9 / 12.3 ✓ | 4.0 / 2.5 / 4.2 | 5.4 / 6.3 / 6.3 |

Table 3: Policy-based role diversity influences the performance of different parameter sharing strategies on the MPE[28] and SMAC[39] benchmarks. The best performance in each scenario is marked ✓. Asterisks denote the algorithms that are not significantly different. Q diversity curves can be referred to Fig. 18.

in table. 3. In small Q diversity ($\simeq 0.5$) scenarios, QMIX significantly outperforms VDN with both shared and no shared strategies. With the increase of Q diversity, the performance of QMIX starts to degrade. In scenarios where agents have significantly different Q value distribution (Fig. 18 1s1m1h1M_vs_3/4/5z), VDN significantly outperforms QMIX. As for IQL, the performance is not as good as VDN and QMIX in most scenarios. However, IQL is not sensitive to Q diversity and can perform well in easy scenarios like 1s1m1h1M_vs_3z. Combined with theoretical analysis in Sec. 4, we can conclude that QMIX is not suitable for a large contribution-based role diversity scenario because of the additional value decomposition module, which is a *sum* function in VDN and a *learnable neural network* in QMIX. The neural network fails to minimize the approximation error for $Q_{tot}$, and is an extra burden when the reward function (or the contribution to the global reward) is diverse. IQL has no such problem as it treats $Q_{tot}$ as the individual Q value. From this part, we conclude that using credit assignment methods with learnable value decomposition module should be avoided in scenarios with large contribution-based role diversity.

## 6 FINDING A BETTER TRAINING STRATEGY

Based on our theoretical and experimental analysis, we contend that role diversity is a strong candidate metric to aid in selecting the proper training strategy for cooperative MARL. Specifically, if the policy-based role diversity is large, we should choose a no parameter sharing strategy, and vice versa. If the trajectory-based role diversity is large, we should avoid communication or other unnecessary information sharing, and vice versa. If the contribution-based role diversity is large, the fixed credit assignment method or independent learning method is preferred, and vice versa. In this way, we can avoid the possible performance bottlenecks in the context of cooperative MARL. Notably, however, the question remains as to how we might obtain an accurate measurement of the role diversity before the training. In this study, all the role diversity comes from VDN and separated training with no communication because we require the baseline method in our study to be robust and training-efficient. The fact however remains that, in some cases, roles are different in ways that are easy to notice and require no trained policy.

## 7 CONCLUSION

In this paper, we define the role and role diversity to describe a cooperative MARL task and explain the question of why the model performance varies in different scenarios. We claim that a strong relationship between the role diversity and model performance exists and we prove it through both theoretical analyses on MARL error bound decomposition and experiments conducted on MARL benchmarks. The experiment results clearly show that the role diversity significantly impacts the model performance of different training strategies and this effect is ubiquitous in various environments and algorithms. Finally, we provide a guideline on choosing the proper training strategies for cooperative MARL based on the role diversity description.

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

## A  PROBLEM FORMULATION

**Multi-agent Reinforcement Learning** A cooperative multi-agent task is a decentralized partially observable Markov decision process [33] with a tuple $G = \langle \mathcal{S}, \mathcal{A}, \mathcal{U}, P, r, \mathcal{Z}, O, n, \gamma \rangle$. Let $\mathcal{S}$ denote the global state of the environment, while $\mathcal{A}$ represents the set of $n$ agents and $\mathcal{U}$ is the action space. At each time step $t$, agent $a \in \mathcal{A} \equiv \{1, ..., n\}$ selects an action $u \in \mathcal{U}$, forming a joint action $\mathbf{u} \in \mathcal{U}^n$, which in turn causes a transition in the environment represented by the state transition function $P(\mathbf{s}'|\mathbf{s}, \mathbf{u}) : \mathcal{S} \times \mathcal{U}^n \times \mathcal{S} \to [0, 1]$. All agents share the same reward function $r(\mathbf{s}, \mathbf{u}) : \mathcal{S} \times \mathcal{U}^n \to \mathbb{R}$ , while $\gamma \in [0, 1)$ is a discount factor. For any state-action pair, the reward $r$ is bounded by $M$, i.e. $|r| \leq M$. We consider a partially observable scenario in which each agent makes individual observations $z \in \mathcal{Z}$ according to the observation function $O(\mathbf{s}, a) : \mathcal{S} \times \mathcal{A} \to \mathcal{Z}$. Each agent has an action-observation history that conditions a stochastic policy $\pi_t$, creating the following joint action value: $Q^\pi(\mathbf{z}_t, \mathbf{u}_t) = \mathbb{E}_{s_{t+1:\infty}, \mathbf{u}_{t+1:\infty}} [R_t|\mathbf{z}_t, \mathbf{u}_t]$, where $R_t = \sum_{i=0}^{\infty} \gamma^i r_{t+i}$ is the discounted return.

**Centralized training with decentralized execution** Centralized training with decentralized execution (CTDE) is a commonly used architecture in the MARL context. Each agent is conditioned only on its own action-observation history to make a decision using the learned policy. The centralized value function provides a centralized gradient to update the individual function based on its output. Therefore, a stronger individual value function can benefit the centralized training.

## B  PROOFS

In this section, we present more detailed results and the proofs for the theoretical analysis in Sec. 4. In Sec. B.1, we denote more notations and state the concentration property of Markov decision process. Sec. B.2 presents two useful lemmas about the error propagation and one-step approximation respectively. In Sec. B.3, we consider a simple example of the decentralized and cooperative MARL and provide the finite-sample analysis for the estimation error of the joint action-value function. We use the value decomposition [35, 48] and the finite-sample results for single-agent RL [10]. For more related results about MARL, please refer to [60] and [53]. Sec. B.4 studies the parameter sharing case that all agents share one deep Q network. In Sec. B.5, we assume $Q$ is a sparse ReLU network and $TQ$ is a composition of Hölder smooth functions. Then we discuss the convergence rate of the statistical error as the sample size tends to infinity. According to Sec. B.3, B.4 and B.5, one can find that each type of role diversity have different impact to the decomposed estimation error. Furthermore, we explain the benefits of the training options, e.g. parameter sharing (Sec. 5.1), communication (Sec. 5.2) and credit assignment (Sec. 5.3), and discuss how these options impact the convergence rate of approximation error and statistical error.

We summarize our results as follows:

- The parameter sharing strategy introduces a bias term by constraining the diversity of individual action-value functions, which corresponds to the policy-based role diversity. At the same time, it speeds up the convergence rate of statistical error by pooling training data.

- The communication mechanism reduces the variance caused by the trajectory-based role diversity but slows down the convergence rate of approximation error by introducing more active input variables.

- When the contribution-based role diversity is nonnegligible, the credit assignment can significantly reduce the estimation error of the action-value function.

### B.1  MORE NOTATIONS AND ASSUMPTIONS

We denote the joint optimal action-value function by

$$Q_{tot}^*(\mathbf{z}, \mathbf{u}) = Q_{tot}^*(z_1, \ldots, z_n, u_1, \ldots, u_n),$$

where $\mathbf{z}$ is the global state of the environment, $\mathbf{u} = \{u_1, \ldots, u_n\}$ is the action set that collects the action of each agent $u_i$ and $z_i$ is the observation of the agent $i$ generated from the emission distribution $z_i \sim \Lambda(z|i, \mathbf{s})$. We further denote the individual optimal action-value function by $Q_i^*$ and write

$$\mathbf{Q}^*(\mathbf{z}, \mathbf{u}) = \left( Q_1^*(z_1, u_1), \ldots, Q_n^*(z_n, u_n) \right)$$

as the vector of all agents' action-value functions. According to the value decomposition assumption, the joint optimal Q-function $Q_{tot}^*$ can be approximated with

$$F(\mathbf{Q}^*)(\mathbf{z}, \mathbf{u}) = F(Q_1^*(\mathbf{z}_1, \mathbf{u}_1), \dots, Q_n^*(\mathbf{z}_n, \mathbf{u}_n)),$$

where $F \in \mathcal{F}$ is a credit assignment function. The VDN method [48] approximates the joint as a sum of individual action-value functions that condition only on individual observations and actions. Then a decentralised policy arises simply from each agent selecting actions greedily with respect to its $Q_i$. Since $n$ is a fixed integer, we write the hypothetical space of credit assign functions that only contains one function as:

$$\mathcal{F} = \big\{ F(\mathbf{Q}) = \frac{1}{n} \sum_{i=1}^n Q_i : \text{ with } \mathbf{Q} = (Q_1, \dots, Q_n) \big\}.$$

The QMIX method [35] generalizes the value decomposition scheme and prove that if

$$\frac{\partial Q_{tot}(\mathbf{z}, \mathbf{u})}{\partial Q_i(\mathbf{z}_i, \mathbf{u}_i)} \geq 0, \text{ for } \forall 1 \leq i \leq n, \ \forall \mathbf{z} \in \mathcal{Z}^n, \ \forall \mathbf{u} \in \mathcal{U}^n,$$

then the global $\arg\max$ performed on joint Q-function yields the same result as a set of individual $\arg\max$ operations performed on each agent Q-function, that is

$$\arg\max_{\mathbf{u}} Q(\mathbf{z}, \mathbf{u}) = \begin{pmatrix} \arg\max_{\mathbf{u}_1} Q_1(\mathbf{z}_1, \mathbf{u}_1) \\ \arg\max_{\mathbf{u}_2} Q_2(\mathbf{z}_2, \mathbf{u}_2) \\ \vdots \\ \arg\max_{\mathbf{u}_n} Q_n(\mathbf{z}_n, \mathbf{u}_n) \end{pmatrix}.$$

Motivated by VDN and QMIX, we consider a simple case throughout this section:

$$\mathcal{F} = \big\{ F(\mathbf{Q}) = \mathbf{w}^\top \mathbf{Q} : \ \mathbf{w} \in \Delta_n \text{ and } \mathbf{Q} = (Q_1, \dots, Q_n) \big\},$$

where $\Delta_n$ is the $n-1$ dimensional probability simplex.

Suppose the individual action-value function is estimated by the fitted-Q iteration (FQI) algorithm [9, 38]. At the iteration $0 \leq t \leq T$, we write $\tilde{Q}_{i,t}$ and $\pi_{i,t}$ as the output of FQI algorithm and the corresponding greedy policy respectively. Let $Q^{\pi_{i,t}}$ be the Q-function corresponding to $\pi_{i,t}$. Then the joint action-value function is estimated by $\hat{F}(\mathbf{Q}_t)$, where

$$\mathbf{Q}_t(\mathbf{z}, \mathbf{u}) = \big( Q^{\pi_{1,t}}(\mathbf{z}_1, \mathbf{u}_1), Q^{\pi_{2,t}}(\mathbf{z}_2, \mathbf{u}_2), \dots, Q^{\pi_{n,t}}(\mathbf{z}_n, \mathbf{u}_n) \big).$$

To proceed further, we give the following assumption that controls the similarity between two probability distributions under the Markov decision process.

**Assumption 1.** *Let $\mu, \nu \in \mathcal{P}(\mathcal{Z} \times \mathcal{U})$ be two probability measures that are absolutely continuous with respect to the Lebesgue measure on $\mathcal{Z} \times \mathcal{U}$. Let $\{\pi_t\}$ be a sequence of joint policies for all the agents, with $\pi_t : \mathcal{Z} \to \mathcal{P}(\mathcal{U})$ for all time $t$. Suppose the initial state-action pair $(\mathbf{z}_0, \mathbf{u}_0)$ has distribution $\mu$, and the action $\mathbf{u}_t$ is sampled from the joint policy $\pi_t$. For any integer $m$, we denote by $P^{\pi_m} P^{\pi_{m-1}} \cdots P^{\pi_1} \mu$ the distribution of $\{(\mathbf{z}_t, \mathbf{u}_t)\}_{t=1}^m$ under the policy sequence $\{\pi_t\}_{t=1,\dots,m}$. Then, the $m$-th concentration coefficient is defined as*

$$\kappa(m; \mu, \nu) = \sup_{\pi_1, \dots, \pi_m} \left[ \mathbb{E}_\nu \Big| \frac{d(P^{\pi_m} P^{\pi_{m-1}} \cdots P^{\pi_1} \mu)}{d\nu} \Big|^2 \right]^{1/2},$$

*where $d(P^{\pi_m} P^{\pi_{m-1}} \cdots P^{\pi_1} \mu)/d\nu$ is the Radon-Nikodym derivative of $P^{\pi_m} P^{\pi_{m-1}} \cdots P^{\pi_1} \mu$ with respect to $\nu$ and the supremum is taken over all possible policies.*

*Furthermore, let $\nu$ be the stationary distribution of the samples $\{(\mathbf{z}_t, \mathbf{u}_t)\}$ from the Markov decision process and let $\mu$ be a fixed distribution on $\mathcal{S} \times \mathcal{U}$. We assume that there exists a constant $\phi_{\mu,\nu}$ such that*

$$(1 - \gamma)^2 \cdot \sum_{m \geq 1} \gamma^{m-1} \cdot m \cdot \kappa(m; \mu, \nu) \leq \phi_{\mu,\nu}.$$

To proceed further, we denote $\mathcal{Q}$ as the space of individual Q-functions and let

$$\omega(\mathcal{Q}) = \sup_{Q \in \mathcal{Q}} \inf_{Q' \in \mathcal{Q}} \|Q' - TQ\|_{2,\nu}^2,$$

where $\| \cdot \|_{2,\nu}$ as the $L_2$ norm with respect to a probability measure $\nu$. In the following, we take $\nu$ as the independent data sampling distribution in the FQI algorithm, e.g. experience replay [26].

We say a collection $\{Q^1, \ldots, Q^K\} \subseteq \mathcal{Q}$ is an $\delta$-cover of $\mathcal{Q}$ if for each $Q \in \mathcal{Q}$, there exists $Q^k$ such that $\|Q - Q^k\| \le \delta$. The $\delta$-covering number of $\mathcal{Q}$ with respect to $\| \cdot \|$ is

$$N(\mathcal{Q}, \delta, \| \cdot \|) := \inf\{K \in \mathbb{N} : \text{there is an } \delta\text{-cover of } \mathcal{Q} \text{ with respect to } \| \cdot \|\}.$$

In the following, we take the $L^\infty(\mathcal{Z} \times \mathcal{U})$ norm on $\mathcal{Q}$ by

$$\|Q - Q'\|_{L^\infty(\mathcal{Z} \times \mathcal{U})} = \sup_{(\mathbf{z}, \mathbf{u}) \in \mathcal{Z} \times \mathcal{U}} |Q(\mathbf{z}, \mathbf{u}) - Q'(\mathbf{z}, \mathbf{u})|.$$

For the sake of simplicity, we rewrite $N(\mathcal{Q}, \delta, \| \cdot \|)$ as $N_\delta$. In Sec. B.3, we study the estimation error of the joint action-value function and prove that the statistical error depends on $\ln N_0/N$, where $N_0$ is the $1/N$-cover of $\mathcal{Q}$ and $N$ is the sample size. For the parameter sharing settings, the sample size increases while the cover number also increases due to the smaller $\delta$. Thus, we still do not know whether the parameter sharing improves the convergence rate of the statistical error. So we present a fine-grain analysis to discuss the convergence rate in Sec B.5.

## B.2 Useful Lemmas

**Lemma 1.** (Theorem 6.1 in [10]). *For each agent $i \in [n]$, we denote $\{\tilde{Q}_{i,t}\}_{0 \le t \le T}$ as the iterates of FQI Algorithm. Let $\pi_{i,t}$ be the one-step greedy policy with respect to $\tilde{Q}_{i,t}$, and let $Q^{\pi_{i,t}}$ be the action-value function corresponding to $\pi_{i,t}$. Under* **Assumption 1***, we have*

$$\|Q_i^* - Q^{\pi_{i,t}}\|_{1,\mu} \le \frac{2\phi_{\mu,\nu}\gamma}{(1-\gamma)^2} \max_{t \in [T]} \|T\tilde{Q}_{i,t-1} - \tilde{Q}_{i,t}\|_{2,\nu} + \frac{4\gamma^{T+1}}{(1-\gamma)^2} M. \tag{7}$$

**Proof:** Please see Appx C.1 of [10] for a complete proof.

$\square$

This lemma quantifies the error propagation procedure of each agent action-value functions through each iteration of FQI Algorithm. The first term on the RHS is the one-step statistical error and will not vanish even when the iteration goes to infinity ($T \to \infty$). For more related error propagation results, please refer to [11, 12, 22, 32, 40].

**Lemma 2.** (Theorem 6.2 in [10]). *Let $\{(z_{ij}, \mathbf{u}_{ij})\}_{j \in [N]}$ be $N$ i.i.d. random variables. For each $j \in [N]$, let $\mathbf{r}_{ij}$ and $z'_{ij}$ be the reward and the next state corresponding to $(z_{ij}, \mathbf{u}_{ij})$. In addition, for any fixed $\tilde{Q}_{i,t-1} \in \mathcal{Q}$, we define $\mathbf{y}_{ij} = \mathbf{r}_{ij} + \gamma \cdot \max_{\mathbf{u}} \tilde{Q}_{i,t-1}(z'_{ij}, \mathbf{u})$. Based on $\{(z_{ij}, \mathbf{u}_{ij}, \mathbf{y}_{ij})\}_{j \in [N]}$, we define $\tilde{Q}_{i,t}$ as*

$$\tilde{Q}_{i,t} = \arg\min_{Q \in \mathcal{Q}} \frac{1}{N} \sum_{j=1}^N \left(Q(z_{ij}, \mathbf{u}_{ij}) - \mathbf{y}_{ij}\right)^2.$$

Then for any $\delta > 0$, we have

$$\|T\tilde{Q}_{i,t-1} - \tilde{Q}_{i,t}\|_{2,\nu}^2 \le 4\omega(\mathcal{Q}) + C\frac{M^2}{(1-\gamma)^2} \frac{\ln N_\delta}{N} + C\frac{M\delta}{1-\gamma}, \tag{8}$$

where $\omega(\mathcal{Q}) = \sup_{Q \in \mathcal{Q}} \inf_{Q' \in \mathcal{Q}} \|Q' - TQ\|_\nu^2$ and $N_\delta$ is the $\delta$-covering number of $\mathcal{Q}$ with respect to the norm $\| \cdot \|_\infty$.

**Proof:** Please see Appx C.2 of [10] for a complete proof.

## B.3 INDIVIDUAL Q-FUNCTION

**Theorem 1**. *We consider the separated strategy that each agent has its own action-value function and reward. All agents' learning process is independent. Suppose $\{\tilde{Q}_{i,t}\}_{0 \leq t \leq T}$ are the output of FQI Algorithm for the agent $i$. Let $\pi_{i,t}$ be the one-step greedy policy with respect to $\tilde{Q}_{i,t}$, and let $Q^{\pi_{i,t}}$ be the action-value function corresponding to $\pi_{i,t}$. We rewrite*

$$\mathbf{Q}_t = \left(Q^{\pi_{1,t}}, Q^{\pi_{2,t}}, \ldots, Q^{\pi_{n,t}}\right), \quad and \quad \mathbf{Q}^* = \left(Q_1^*, Q_2^*, \ldots, Q_n^*\right).$$

*Recall that $0 \leq \gamma < 1$ is the discount factor, the reward function is bounded, i.e., $|r(\mathbf{s}, \mathbf{u})| \leq M$, $\mathcal{Q}$ is the space of individual Q-functions and $\omega(\mathcal{Q}) = \sup_{Q \in \mathcal{Q}} \inf_{Q' \in \mathcal{Q}} \|Q' - TQ\|_{2,\nu}^2$. Then, under* **Assumption 1**, *we have*

$$
\begin{aligned}
\|Q_{tot}^* - \hat{F}(\mathbf{Q}_t)\|_{1,\mu} \quad \leq \quad & \|Q_{tot}^* - (\mathbf{w}^*)^\top \mathbf{Q}^*\|_{1,\mu} \\
& + \sqrt{n} \times \|\mathbf{w}^* - \hat{\mathbf{w}}\| \times \left\|\sqrt{\mathrm{Var}_n\left(\mathbf{Q}^*(\mathbf{z}, \mathbf{u})\right)}\right\|_{1,\mu} \\
& + \frac{4\phi_{\mu,\nu}\gamma}{(1-\gamma)^2}\sqrt{\omega(\mathcal{Q})} + O\left(\sqrt{\frac{\ln N_0}{N}}\right) + \frac{4\gamma^{T+1}}{(1-\gamma)^2}M,
\end{aligned}
$$

*where $N_0$ is the $1/N$-covering number of $\mathcal{Q}$ with respect to the norm $\|\cdot\|_\infty$ and*

$$\mathrm{Var}_n\left(\mathbf{Q}^*(\mathbf{z}, \mathbf{u})\right) = \frac{1}{n}\sum_{i=1}^n \left(Q_i^*(\mathbf{z}_i, \mathbf{u}_i) - \frac{1}{n}\sum_{i=1}^n Q_i^*(\mathbf{z}_i, \mathbf{u}_i)\right)^2.$$

**Proof**: It is easy to see that

$$
\begin{aligned}
& \|Q_{tot}^* - \hat{F}(\mathbf{Q}_t)\|_{1,\mu} \\
= \quad & \|Q_{tot}^* - F^*(\mathbf{Q}^*) + F^*(\mathbf{Q}^*) - \hat{F}(\mathbf{Q}^*) + \hat{F}(\mathbf{Q}^*) - \hat{F}(\mathbf{Q}_t)\|_{1,\mu} \\
\leq \quad & \|Q_{tot}^* - F^*(\mathbf{Q}^*)\|_{1,\mu} + \|F^*(\mathbf{Q}^*) - \hat{F}(\mathbf{Q}^*)\|_{1,\mu} + \|\hat{F}(\mathbf{Q}^*) - \hat{F}(\mathbf{Q}_t)\|_{1,\mu}. \quad (9)
\end{aligned}
$$

Here the first term at the RHS of (9):

$$\|Q_{tot}^* - F^*(\mathbf{Q}^*)\|_{1,\mu} = \|Q_{tot}^* - (\mathbf{w}^*)^\top \mathbf{Q}^*\|_{1,\mu} \quad (10)$$

represents the best achievable estimation error under the value decomposition assumption. Next we consider the second term in the inequality (9):

$$
\begin{aligned}
\|F^*(\mathbf{Q}^*) - \hat{F}(\mathbf{Q}^*)\|_{1,\mu} \quad = \quad & \|(\mathbf{w}^*)^\top \mathbf{Q}^* - \hat{\mathbf{w}}^\top \mathbf{Q}^*\|_{1,\mu} \\
= \quad & \|(\mathbf{w}^* - \hat{\mathbf{w}})^\top \mathbf{Q}^*\|_{1,\mu}
\end{aligned}
$$

For any given $\mathbf{z} = (\mathbf{z}_1, \ldots, \mathbf{z}_n)$ and $\mathbf{u} = (\mathbf{u}_1, \ldots, \mathbf{u}_n)$,

$$
\begin{aligned}
(\mathbf{w}^* - \hat{\mathbf{w}})^\top \mathbf{Q}^*(\mathbf{z}, \mathbf{u}) \quad = \quad & \sum_{i=1}^n (\mathbf{w}_i^* - \hat{\mathbf{w}}_i) Q_i^*(\mathbf{z}_i, \mathbf{u}_i) \\
= \quad & \sum_{i=1}^n (\mathbf{w}_i^* - \hat{\mathbf{w}}_i)\left(Q_i^*(\mathbf{z}_i, \mathbf{u}_i) - \frac{1}{n}\sum_{i=1}^n Q_i^*(\mathbf{z}_i, \mathbf{u}_i)\right).
\end{aligned}
$$

The second equality holds since $\sum_{i=1}^n (\mathbf{w}_i^* - \hat{\mathbf{w}}_i) \times c = 0$ for any constant $c$. By the Cauchy–Schwarz inequality, we have

$$
\begin{aligned}
(\mathbf{w}^* - \hat{\mathbf{w}})^\top \mathbf{Q}^*(\mathbf{z}, \mathbf{u}) \quad \leq \quad & \sqrt{\sum_{i=1}^n (\mathbf{w}_i^* - \hat{\mathbf{w}}_i)^2} \times \sqrt{\sum_{i=1}^n \left(Q_i^*(\mathbf{z}_i, \mathbf{u}_i) - \frac{1}{n}\sum_{i=1}^n Q_i^*(\mathbf{z}_i, \mathbf{u}_i)\right)^2} \\
= \quad & \|\mathbf{w}^* - \hat{\mathbf{w}}\| \times \sqrt{n \mathrm{Var}_n\left(\mathbf{Q}^*(\mathbf{z}, \mathbf{u})\right)}
\end{aligned}
$$

where

$$\mathrm{Var}_n\left(\mathbf{Q}^*(\mathbf{z}, \mathbf{u})\right) = \frac{1}{n}\sum_{i=1}^n \left(Q_i^*(\mathbf{z}_i, \mathbf{u}_i) - \frac{1}{n}\sum_{i=1}^n Q_i^*(\mathbf{z}_i, \mathbf{u}_i)\right)^2$$

is the variance of the output vector of $\mathbf{Q}^*$ given $\mathbf{z}$ and $\mathbf{u}$. Plugging the positive upper boundary of $(\mathbf{w}^* - \hat{\mathbf{w}})^\top \mathbf{Q}^*$ into the expression of $\|F^*(\mathbf{Q}^*) - \hat{F}(\mathbf{Q}^*)\|_{1,\mu}$, we obtain that

$$
\begin{aligned}
\|F^*(\mathbf{Q}^*) - \hat{F}(\mathbf{Q}^*)\|_{1,\mu} &\leq \left\| \|\mathbf{w}^* - \hat{\mathbf{w}}\| \cdot \sqrt{n \cdot \mathrm{Var}_n\big(\mathbf{Q}^*(\mathbf{z}, \mathbf{u})\big)} \right\|_{1,\mu} \\
&= \sqrt{n} \times \|\mathbf{w}^* - \hat{\mathbf{w}}\| \times \left\| \sqrt{\mathrm{Var}_n\big(\mathbf{Q}^*(\mathbf{z}, \mathbf{u})\big)} \right\|_{1,\mu}. \quad (11)
\end{aligned}
$$

Here the term $\|\sqrt{\mathrm{Var}_n\big(\mathbf{Q}^*(\mathbf{z}, \mathbf{u})\big)}\|_{1,\mu}$ stands for the diversity of the agents.

Finally, we deal with the third term on the RHS of (9). Notice that

$$
\begin{aligned}
\|\hat{F}(\mathbf{Q}^*) - \hat{F}(\mathbf{Q}_t)\|_{1,\mu} &= \|\sum_{i=1}^n \hat{w}_i (Q_i^* - Q_i^{\pi_t})\|_{1,\mu} \\
&\leq \sum_{i=1}^n \hat{w}_i \|Q_i^* - Q_i^{\pi_t}\|_{1,\mu}.
\end{aligned}
$$

Therefore, it suffices to consider the deep Q-learning procedure of each agent separately and to give upper bound of $\|Q_i^* - Q_i^{\pi_t}\|_{1,\mu}$ for each $i \in [n]$. According to (7) and (8),

$$
\begin{aligned}
\|Q_i^* - Q_i^{\pi_t}\|_{1,\mu} &\leq \frac{2\phi_{\mu,\nu}\gamma}{(1-\gamma)^2} \max_{t \in [T]} \|T\tilde{Q}_{i,t-1} - \tilde{Q}_{i,t}\|_{2,\nu} + \frac{4\gamma^{T+1}}{(1-\gamma)^2} M \\
&\leq \frac{2\phi_{\mu,\nu}\gamma}{(1-\gamma)^2} \sqrt{4\omega(\mathcal{Q}) + C\frac{M^2}{(1-\gamma)^2}\frac{\ln N_\delta}{n} + C'\frac{M\delta}{1-\gamma}} + \frac{4\gamma^{T+1}}{(1-\gamma)^2} M \\
&\leq \frac{4\phi_{\mu,\nu}\gamma}{(1-\gamma)^2} \sqrt{\omega(\mathcal{Q})} + C\frac{2M\phi_{\mu,\nu}\gamma}{(1-\gamma)^3}\sqrt{\frac{\ln N_\delta}{N}} + C'\frac{2\sqrt{M}\phi_{\mu,\nu}\gamma}{(1-\gamma)^{5/2}}\sqrt{\delta} + \frac{4\gamma^{T+1}}{(1-\gamma)^2} M.
\end{aligned}
$$

We take $\delta = 1/N$ and write $N_0$ as the $1/N$-covering number of $\mathcal{Q}$. Then, we have

$$
\|Q_i^* - Q_i^{\pi_t}\|_{1,\mu} \leq \frac{4\phi_{\mu,\nu}\gamma}{(1-\gamma)^2}\sqrt{\omega(\mathcal{Q})} + O\big(\sqrt{\frac{\ln N_0}{N}}\big) + \frac{4\gamma^{T+1}}{(1-\gamma)^2} M.
$$

Furthermore,

$$
\|\hat{F}(\mathbf{Q}^*) - \hat{F}(\mathbf{Q}_t)\|_{1,\mu} \leq \frac{4\phi_{\mu,\nu}\gamma}{(1-\gamma)^2}\sqrt{\omega(\mathcal{Q})} + O\big(\sqrt{\frac{\ln N_0}{N}}\big) + \frac{4\gamma^{T+1}}{(1-\gamma)^2} M. \quad (12)
$$

Combining the results of (10), (11) and (12), we know

$$
\begin{aligned}
\|Q_{tot}^* - \hat{F}(\mathbf{Q}_t)\|_{1,\mu} &\leq \|Q_{tot}^* - (\mathbf{w}^*)^\top \mathbf{Q}^*\|_{1,\mu} \\
&\quad + \sqrt{n} \times \|\mathbf{w}^* - \hat{\mathbf{w}}\| \times \left\| \sqrt{\mathrm{Var}_n\big(\mathbf{Q}^*(\mathbf{z}, \mathbf{u})\big)} \right\|_{1,\mu} \\
&\quad + \frac{4\phi_{\mu,\nu}\gamma}{(1-\gamma)^2}\sqrt{\omega(\mathcal{Q})} + O\big(\sqrt{\frac{\ln N_0}{N}}\big) + \frac{4\gamma^{T+1}}{(1-\gamma)^2} M.
\end{aligned}
$$

$\square$

**Remark.** The term

$$
\sqrt{n} \times \|\mathbf{w}^* - \hat{\mathbf{w}}\| \times \left\| \sqrt{\mathrm{Var}_n\big(\mathbf{Q}^*(\mathbf{z}, \mathbf{u})\big)} \right\|_{1,\mu}
$$

shows the benefits of learning credit assignment, where $\mathbf{w}^*$ stands for the best credit assignment scheme. Here we assume $\hat{\mathbf{w}}$ is given and do not take the modelling and learning of $\hat{\mathbf{w}}$ into consideration. In practice, $\hat{\mathbf{w}}$ is the output of a credit distribution network and its learning procedure also influence the convergence properties of individual Q-functions. On the other hand, $\mathrm{Var}_n(\mathbf{Q})$ corresponds to the contribution-based role diversity in Sec. 3.3. Therefore, when the variance is nonzero, a good credit assignment $\hat{\mathbf{w}}$ can the estimation error. For the parameter sharing case in the next section, we decompose the variance term into the sum of a bias and a variance caused by policy-based role diversity and the trajectory-based role diversity respectively. This decomposition does not always hold. Here, we assume that all agents' learning processes are independent and that each agent has its own reward function. In practice, these assumptions are idealistic and limited.

### B.4 SHARED Q-FUNCTION

**Theorem 2**. *We consider the parameter sharing strategy that all individual agents shares one action-value function. Suppose $\{\tilde{Q}_t\}_{0 \leq t \leq T}$ are the output of FQI Algorithm. Let $\pi_t$ be the one-step greedy policy with respect to $\tilde{Q}_t$, and let $Q^{\pi_t}$ be the action-value function corresponding to $\pi_t$. We further denote $\bar{Q}^*$ as the optimal shared action-value function and write*

$$\bar{\mathbf{Q}}_t(\mathbf{z}, \mathbf{u}) = \left( Q^{\pi_t}(\mathbf{z}_1, \mathbf{u}_1), Q^{\pi_t}(\mathbf{z}_2, \mathbf{u}_2), \ldots, Q^{\pi_t}(\mathbf{z}_n, \mathbf{u}_n) \right),$$
$$\bar{\mathbf{Q}}^*(\mathbf{z}, \mathbf{u}) = \left( \bar{Q}^*(\mathbf{z}_1, \mathbf{u}_1), \bar{Q}^*(\mathbf{z}_2, \mathbf{u}_2), \ldots, \bar{Q}^*(\mathbf{z}_n, \mathbf{u}_n) \right).$$

*Recall that $0 \leq \gamma < 1$ is the discount factor, the reward function is bounded, i.e., $|r(\mathbf{s}, \mathbf{u})| \leq M$, $\mathcal{Q}$ is the space of individual Q-functions and $\omega(\mathcal{Q}) = \sup_{Q \in \mathcal{Q}} \inf_{Q' \in \mathcal{Q}} \|Q' - TQ\|_{2,\nu}^2$. Then, under* **Assumption 1**, *we have*

$$\|Q_{tot}^* - \hat{F}(\bar{\mathbf{Q}}_t)\|_{1,\mu} \leq \|Q_{tot}^* - (\mathbf{w}^*)^\top \mathbf{Q}^*\|_{1,\mu} + \left\| \sum_{i=1}^n \mathbf{w}_i^*(Q_i^* - \bar{Q}^*) \right\|_{1,\mu}$$
$$+ \sqrt{n} \times \|\mathbf{w}^* - \hat{\mathbf{w}}\| \times \left\| \sqrt{\mathrm{Var}_n\big(\bar{\mathbf{Q}}^*(\mathbf{z}, \mathbf{u})\big)} \right\|_{1,\mu}$$
$$+ \frac{4\phi_{\mu,\nu}\gamma}{(1-\gamma)^2}\sqrt{\omega(\mathcal{Q})} + O\left(\sqrt{\frac{\ln N_0'}{nN}}\right) + \frac{4\gamma^{T+1}}{(1-\gamma)^2}M.$$

*where $N_0'$ is the $1/(nN)$-covering number of $\mathcal{Q}$ with respect to the norm $\|\cdot\|_\infty$ and*

$$\mathrm{Var}_n\big(\bar{\mathbf{Q}}^*(\mathbf{z}, \mathbf{u})\big) = \frac{1}{n}\sum_{i=1}^n \left( \bar{Q}^*(\mathbf{z}_i, \mathbf{u}_i) - \frac{1}{n}\sum_{i=1}^n \bar{Q}^*(\mathbf{z}_i, \mathbf{u}_i) \right)^2.$$

**Proof**: Similar to the arguments in (9), we have

$$\|Q_{tot}^* - \hat{F}(\bar{\mathbf{Q}}_t)\|_{1,\mu} \leq \|Q_{tot}^* - F^*(\mathbf{Q}^*)\|_{1,\mu} + \|F^*(\mathbf{Q}^*) - F^*(\bar{\mathbf{Q}}^*)\|_{1,\mu}$$
$$+ \|F^*(\bar{\mathbf{Q}}^*) - \hat{F}(\bar{\mathbf{Q}}^*)\|_{1,\mu} + \|\hat{F}(\bar{\mathbf{Q}}^*) - \hat{F}(\bar{\mathbf{Q}}_t)\|_{1,\mu}. \quad (13)$$

The term $\|Q_{tot}^* - F^*(\mathbf{Q}^*)\|_{1,\mu}$ caused by the value decomposition is the same to that in (9). So (10) still holds. For the second term on the RHS of (13),

$$\|F^*(\mathbf{Q}^*) - F^*(\bar{\mathbf{Q}}^*)\|_{1,\mu} = \|(\mathbf{w}^*)^\top(\mathbf{Q}^* - \bar{\mathbf{Q}}^*)\|_{1,\mu}$$
$$= \left\| \sum_{i=1}^n \mathbf{w}_i^*(Q_i^* - \bar{Q}^*) \right\|_{1,\mu},$$

which is the bias term caused by the parameter sharing. Next, we turn to a turn that is related to the trajectory-based role diversity. Similar to (11), we know

$$\big(F^*(\bar{\mathbf{Q}}^*) - \hat{F}(\bar{\mathbf{Q}}^*)\big)(\mathbf{z}, \mathbf{u}) = (\mathbf{w}^* - \hat{\mathbf{w}})^\top \bar{\mathbf{Q}}^*(\mathbf{z}, \mathbf{u})$$
$$= \sum_{i=1}^n (\mathbf{w}_i^* - \hat{\mathbf{w}}_i)\bar{Q}^*(\mathbf{z}_i, \mathbf{u}_i)$$
$$= \sum_{i=1}^n (\mathbf{w}_i^* - \hat{\mathbf{w}}_i)\left( \bar{Q}^*(\mathbf{z}_i, \mathbf{u}_i) - \frac{1}{n}\sum_{i=1}^n \bar{Q}^*(\mathbf{z}_i, \mathbf{u}_i) \right)$$
$$= \sqrt{n} \times \|\mathbf{w}^* - \hat{\mathbf{w}}\| \times \sqrt{\mathrm{Var}_n\big(\bar{\mathbf{Q}}^*(\mathbf{z}, \mathbf{u})\big)}.$$

On the other hand, by (7) and (8),

$$\|Q^* - Q^{\pi_t}\|_{1,\mu} \leq \frac{2\phi_{\mu,\nu}\gamma}{(1-\gamma)^2}\max_{t \in [T]}\|T\tilde{Q}_{t-1} - \tilde{Q}_t\|_{2,\nu} + \frac{4\gamma^{T+1}}{(1-\gamma)^2}M$$
$$\leq \frac{2\phi_{\mu,\nu}\gamma}{(1-\gamma)^2}\sqrt{4\omega(\mathcal{Q}) + C\frac{M^2}{(1-\gamma)^2}\frac{\ln N_\delta}{nN} + C'\frac{M\delta}{1-\gamma}} + \frac{4\gamma^{T+1}}{(1-\gamma)^2}M$$
$$\leq \frac{4\phi_{\mu,\nu}\gamma}{(1-\gamma)^2}\sqrt{\omega(\mathcal{Q})} + C\frac{2M\phi_{\mu,\nu}\gamma}{(1-\gamma)^3}\sqrt{\frac{\ln N_\delta}{nN}} + C'\frac{2\sqrt{M}\phi_{\mu,\nu}\gamma}{(1-\gamma)^{5/2}}\sqrt{\delta} + \frac{4\gamma^{T+1}}{(1-\gamma)^2}M.$$

We take $\delta = 1/(nN)$ and write $N_0'$ as the $1/(nN)$-covering number of $\mathcal{Q}$. Then, we have

$$\|Q^* - Q^{\pi_t}\|_{1,\mu} \leq \frac{4\phi_{\mu,\nu}\gamma}{(1-\gamma)^2}\sqrt{\omega(\mathcal{Q})} + O(\sqrt{\frac{\ln N_0'}{nN}}) + \frac{4\gamma^{T+1}}{(1-\gamma)^2}M.$$

Therefore,

$$\|\hat{F}(\bar{\mathbf{Q}}^*) - \hat{F}(\bar{\mathbf{Q}}_t)\|_{1,\mu} \leq \frac{4\phi_{\mu,\nu}\gamma}{(1-\gamma)^2}\sqrt{\omega(\mathcal{Q})} + O(\sqrt{\frac{\ln N_0'}{nN}}) + \frac{4\gamma^{T+1}}{(1-\gamma)^2}M.$$

Summarizing the above results, we have

$$\|Q_{tot}^* - \hat{F}(\bar{\mathbf{Q}}_t)\|_{1,\mu} \leq \|Q_{tot}^* - (\mathbf{w}^*)^\top \mathbf{Q}^*\|_{1,\mu} + \left\|\sum_{i=1}^n \mathbf{w}_i^*(Q_i^* - \bar{Q}^*)\right\|_{1,\mu}$$

$$+\sqrt{n} \times \|\mathbf{w}^* - \hat{\mathbf{w}}\| \times \left\|\sqrt{\mathrm{Var}_n(\bar{\mathbf{Q}}^*(\mathbf{z},\mathbf{u}))}\right\|_{1,\mu}$$

$$+\frac{4\phi_{\mu,\nu}\gamma}{(1-\gamma)^2}\sqrt{\omega(\mathcal{Q})} + O(\sqrt{\frac{\ln N_0'}{nN}}) + \frac{4\gamma^{T+1}}{(1-\gamma)^2}M.$$

$\square$

## B.5 Convergence Rate

Similar to the Theorem 4.4 in [10], we assume that $Q$ belongs to a family of sparse ReLU networks and $TQ$ can be written as compositions of Hölder smooth functions. Here $T$ is the optimal Bellman operator. We start with the definition of a $(L+1)$ layers and $\{d_j\}_{j=1}^{L+1}$ width ReLU networks:

$$f(x) = W_{L+1}\sigma(W_L\sigma(W_{L-1}\ldots\sigma(W_2\sigma(W_1 x + v_1) + v_2)\ldots v_{L-1}) + v_L),$$

where $\sigma$ is the ReLU activation function, and $W_l$ and $v_l$ are the weight matrix and the bias in the $l$-th layer, respectively. The family of sparse ReLU networks is defined as

$$\mathcal{F}(L, \{d_j\}_{i=0}^{L+1}, s) = \left\{f : \max_{l\in[L+1]}\|\tilde{W}_l\|_\infty \leq 1, \sum_{l=1}^{L+1}\|\tilde{W}_l\|_0 \leq s, \max_{j\in[d_{L+1}]}\|f_j\|_\infty \leq \frac{M}{1-\gamma}\right\},$$

where $\tilde{W}_l$ is the parameter matrix that contains $W_l$ and $v_l$ and $f_j$ is the $j$-th output of $f$. On the other hand, the set of Hölder smooth functions is

$$\mathcal{C}_r(\mathcal{D},\beta,H) = \left\{f : \mathcal{D} \to \mathbb{R} : \sum_{\boldsymbol{\alpha}:|\boldsymbol{\alpha}|<\beta}\|\partial^{\boldsymbol{\alpha}}f\|_\infty + \sum_{\boldsymbol{\alpha}:\|\boldsymbol{\alpha}\|_1=\lfloor\beta\rfloor}\sup_{\substack{x,y\in\mathcal{D},\\x\neq y}}\frac{|\partial^{\boldsymbol{\alpha}}f(x) - \partial^{\boldsymbol{\alpha}}f(y)|}{\|x-y\|_\infty^{\beta-\lfloor\beta\rfloor}} \leq H\right\},$$

where $\mathcal{D}$ is a a compact subset of $\mathbb{R}^r$, $\lfloor\cdot\rfloor$ stands for the floor function and $\partial^{\boldsymbol{\alpha}} = \partial^{\alpha_1}\partial^{\alpha_2}\cdots\partial^{\alpha_r}$ with $\boldsymbol{\alpha} = (\alpha_1,\alpha_2,\ldots,\alpha_n)^\top$. Furthermore, we write $\mathcal{G}(\{p_j,t_j,\beta_j,H_j\}_{j\in[q]})$ as the family of functions that can be decomposed into the composition of a sequence of Hölder smooth functions $\{g_j\}_{j\in[q]}$. That is, for any function $f \in \mathcal{G}(\{p_j,t_j,\beta_j,H_j\}_{j\in[q]})$,

$$f = g_q \circ g_{q-1} \circ \cdots \circ g_2 \circ g_1,$$

where for any $k \in [p_{j+1}]$ and $j \in [q]$, the $k$-th component of the function $g_j$ is a Hölder smooth function, i.e., $g_{jk} \in \mathcal{C}_{t_j}([a_j,b_j]^{t_j},\beta_j,H_j)$. For simplicity, we take $p_{j+1} = 1$. Here we assume that the input of $g_{jk}$ is $t_j$-dimensional, where $t_j$ can be much smaller than $p_j$. More specific, the deep Q network we used is a sparse ReLU network for any given action u. Therefore, we rewrite the space of individual Q-functions $\mathcal{Q}$ as

$$\mathcal{F}_0 = \{f : S \times U \to \mathbb{R} : f(,\mathbf{u}) \in \mathcal{F}(L,\{d_j\}_{j=0}^{L+1}, s) \text{ for any } \mathbf{u} \in U\}.$$

Furthermore, for any $Q \in \mathcal{F}_0$, we assume $TQ$ is a composition of Hölder smooth functions and belongs to the following family:

$$\mathcal{G}_0 = \{f : S \times U \to \mathbb{R} : f(,\mathbf{u}) \in \mathcal{G}(\{p_j,t_j,\beta_j,H_j\}_{j\in[q]}) \text{ for any } \mathbf{u} \in U\}.$$

To proceed further, we denote

$$\alpha^* = \max_{j \in [q]} \frac{t_j}{2\beta_j^* + t_j}, \quad \beta_j^* = \beta_j \times \prod_{l=j+1} \min(\beta_l, 1), \quad \text{and} \quad \beta_q^* = 1.$$

Now we are ready to state the following result.

**Theorem 3**. *Suppose the assumptions of Theorem 1 hold and for any $Q \in \mathcal{F}_0$, $TQ \in \mathcal{G}_0$, where $T$ is the optimal Bellman operator. The sample size $N$ is sufficiently large such that there exists a constant $\xi > 0$ satisfies*

$$\max \left\{ \sum_{j=1}^{q} (t_j + \beta_j + 1)^{3+t_j}, \sum_{j \in [q]} \ln(t_j + \beta_j), \max_{j \in [q]} p_j \right\} \lesssim (\ln N)^{\xi}.$$

*The network architecture of the Q-function is well designed such that*

$$L \lesssim (\ln N)^{\xi^*}, \quad r \leq \min_{j \in [L]} d_j \leq \max_{j \in [L]} d_j \lesssim N^{\xi^*}, \quad \text{and} \quad s \asymp N^{\alpha^*} (\ln N)^{\xi^*}$$

*for some constant $\xi^* > 1 + 2\xi$. The number of iterations $T$ is sufficiently large, such that*

$$T \geq C'(1 - \alpha^*) \ln N,$$

*where $C'$ is a constant. Then, under **Assumption 1**, we have*

$$
\begin{aligned}
\|Q_{tot}^* - \hat{F}(\mathbf{Q}_t)\|_{1,\mu} \leq{}& \|Q_{tot}^* - (\mathbf{w}^*)^\top \mathbf{Q}^*\|_{1,\mu} \\
&+ \sqrt{n} \times \|\mathbf{w}^* - \hat{\mathbf{w}}\| \times \left\| \sqrt{\mathrm{Var}_n(\mathbf{Q}^*(\mathbf{z}, \mathbf{u}))} \right\|_{1,\mu} \\
&+ O\left( (\ln N)^{1+2\xi^*} N^{-\min_{j \in [q]} \frac{\beta_j^*}{2\beta_j^* + t_j}} \right).
\end{aligned}
$$

**Proof:** This is a direct conclusion reached by Theorem 4.4 in [10]. That is, for any agent $i \in [n]$,

$$\|Q_i^* - Q^{\pi_{i,t}}\|_{1,\mu} \leq O\left( (\ln N)^{1+2\xi^*} N^{(\alpha^*-1)/2} \right) + \frac{4\gamma^{T+1}}{(1-\gamma)^2} M.$$

The approximation error in Theorem 1 that involves $\omega(\mathcal{Q})$ is bounded above via Theorem 5 in [41]. The upper bound for the cover number $N_0$ is derived from Theorem 14.5 in [2]. Please refer to Section 6 in [10] and Sec. B.3 for a complete proof. $\qquad \square$

**Remark:** Note that

$$\frac{4\gamma^{T+1}}{(1-\gamma)^2} M \to 0 \quad \text{as} \quad T \to \infty,$$

which is the algorithmic error that converges to zero at a linear rate of $T$. In Theorem 3, we assume $T$ is sufficiently large such that this error is negligible comparing to the statistical error. If we ignore the logarithmic term, the convergence rate of the statistical error is about

$$\max_{j \in [q]} N^{-\frac{\beta_j^*}{2\beta_j^* + t_j}}.$$

Here $\beta_j^*$ and $t_j$ are parameters of the functional space of $TQ$. Therefore, the parameter sharing (Sec. 5.1) keeps $\beta_j^*$ and $t_j$ unchanged and increases the sample size $N$ to $nN$ by pooling training data. In addition, $t_j$ is the number of active input variables of $g_j$. Thus, the communication mechanism (Sec. 5.2) slows down the convergence rate by enlarging $t_j$.

## C    OBSERVATION OVERLAP PERCENTAGE CALCULATION

### C.1    OVERLAP PERCENTAGE CALCULATION IN GAMES

In this part, we demonstrate how to calculate the observation overlap percentage in SMAC [39]. As the partial observable area is circular, and the coordinate system is a 2D map with axis X and Y, the observation overlap on one battle scenario can be computed as:

$$
\begin{aligned}
l &= \sqrt{(x_{a_0} - x_{a_1}) \cdot (x_{a_1} - x_{a_0}) + (y_{a_0} - y_{a_1}) \cdot (y_{a_1} - y_{a_0})} \\
p &= (1 + 2 \cdot r)/2 \\
s &= 2 \cdot \sqrt{p \cdot (p - l) \cdot (p - r) \cdot (p - r)} \\
o &= 2 \cdot cos^{-1}(l/(2 \cdot r)) \cdot r \cdot r - s \\
d_T^{a_0, a_1} &= o/(\pi \cdot r^2)
\end{aligned}
\tag{14}
$$

Here $r$ is the vision scope. Notice that if $l < 2r$, $d_T$ equals zero as no overlap exists. We provide the observation overlap curve in Fig. 2b to show how trajectory-based role distance varies in one game.

### C.2    OVERLAP PERCENTAGE CALCULATION IN REAL WORLD SCENARIO (SEMANTIC)

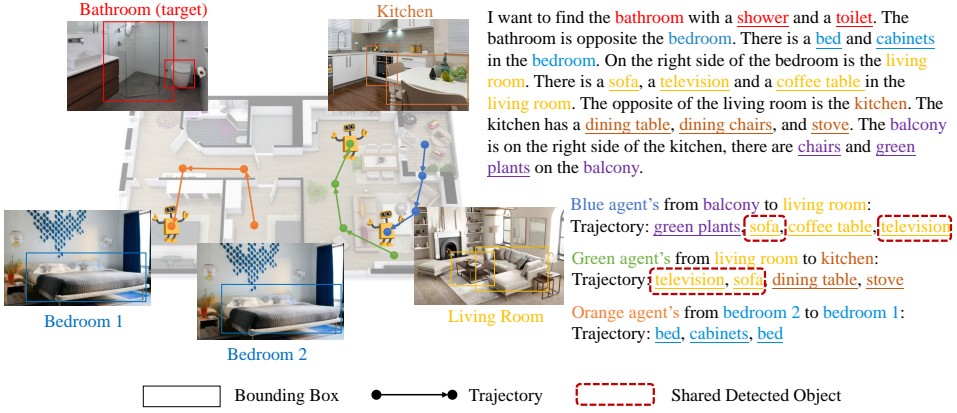

Figure 5: A multi-agent visual language navigation task. Agents are initialized in different locations and the target description is given. Agents need to cooperate with each other to find the target location according to the description as soon as possible.

In this part, we demonstrate how to apply the observation overlap concept and trajectory-based role diversity calculation (Eq. 3) to real-world scenarios. Different from game scenarios like SMAC and MPE, the observation of real-world tasks is usually an image. For example, in the vision language navigation task (VLN[1]), agents take real indoor scene pictures as the input, combine them with language description to locate the target as shown in Fig. 5. Considering the learning efficiency, object detection techniques like YOLO[36] and FasterRCNN[37] are used in VLN to help extract the objects from the scene pictures as semantic information. The semantic information can be recorded as part of the agents' trajectory, enabling agents to use the past information for future decisions. Under the multi-agent setting, agents are required to cooperate and find the target together. Therefore, trajectory overlap should be avoided, which means that large trajectory-based role diversity is preferred in this task, and policies that cause trajectory overlap should be punished. Directly using scene pictures as input or its feature pattern can bring large noise in observation overlap calculation. Instead, using semantic information from the detected object can significantly reduce the noise and serve as a good observation history representation. As shown in Fig. 5, the red dotted frames indicate that the blue agent and green agent share some similar observation semantic in their trajectories. In this way, the trajectory-based role diversity of multi-agent VLN task can be calculated the same as Eq. 3. Only the $d_T^{a_0, a_1}$ is replaced by the observation semantic overlap, which is the shared detected object percentage in total detected objects. In this way, without knowing the exact trajectory, we still manage to calculate the trajectory distance. And using overlap to represent the trajectory-based role diversity, we can keep this metric in a fixed range from 0 to 1.

## C.3 Overlap Percentage Calculation in Real World Scenario (Raw)

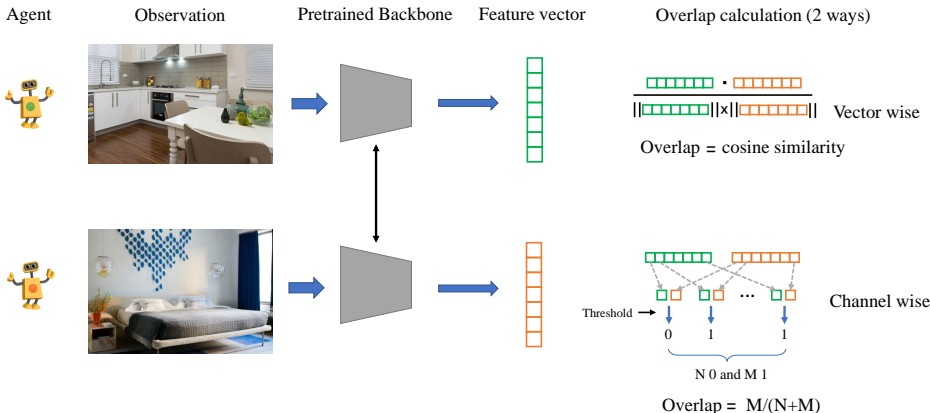

Figure 6: Using real images as observation overlap percentage calculation. Two methods are proposed including vector-wise cosine similarity and channel-wise threshold-based similarity percentage. A detailed discussion can be found in Sec. C.3

It is also possible that we can get observation overlap directly based on real image MARL tasks. As showed in Fig. 6. Passing the input image to pre-trained CNN/Transformer backbone and getting its feature, we can use cosine similarity or channel-wise similarity to compute the overlap between different observation features as $d_T^{a_0, a_1}$ in Eq. 3. However, these methods can bring large noise to this metric. Moreover, how to stabilize the reinforcement learning with real pictures as input is still under investigation. In addition, it is rare in MARL tasks that the only information provided in the training stage is one single image. Location and communication are necessary auxiliary information to help learn the coordination of agents in most MARL tasks. Therefore, simply using the raw image to calculate the observation overlap can be a choice, but not the best choice.

## D Types of Role

We present two illustration figures for different types of role based on MPE [28] and SMAC [39]. Fig. 7 is based on MPE. Grey circles and black circles represent agents and goals respectively. Dashed arrows in different colors represent different actions. Larger circles receive more rewards when they reach the goal.

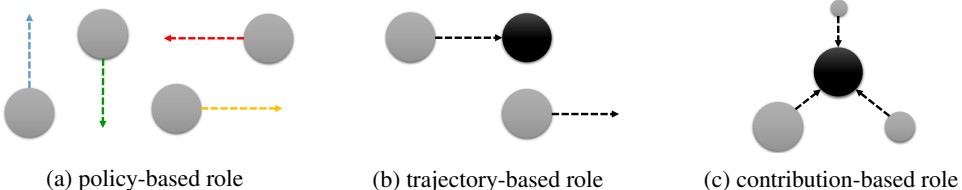

(a) policy-based role    (b) trajectory-based role    (c) contribution-based role

Figure 7: An illustration of different role based on MPE.

Fig. 8 is based on SMAC. A detailed explanation can be found in the caption.

## E Connections of Different Roles

Is there any redundancy in the definition of different kinds of role diversity in Sec. 3? Here we discuss the connections of different role diversity.

From the theoretical perspective, the contribution-based role diversity is a compound description of role diversity. It corresponds to the variance term in (5). For the parameter sharing case, we decompose this variance into a sum of two terms: a bias term corresponds to the policy-based role

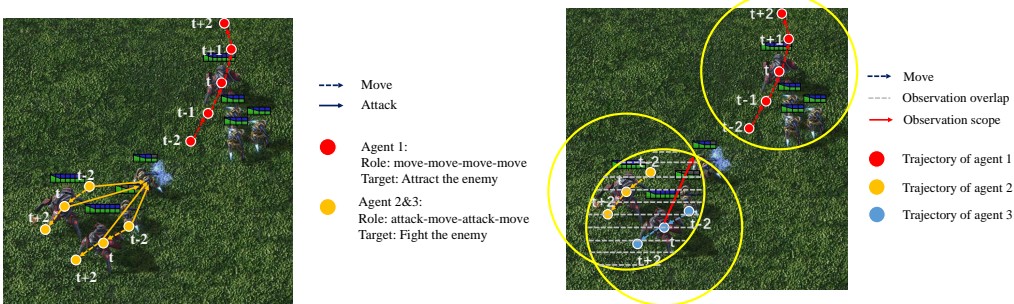

(a) Policy-based role difference in a period from t-2 to t+2. Action statistic shows only two different roles among three agents.

(b) Trajectory-based role difference in a period from t-2 to t+2. The area covered by the grey dotted line is the observation overlap of agents 2 and 3. There is no overlap for agent 1.

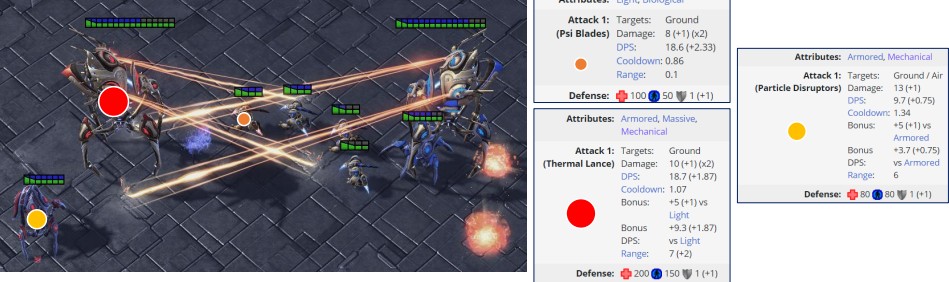

(c) Contribution-based role difference depends on agents' original attributes including attack method and defense.

Figure 8: Illustration of policy-based role, trajectory-based role, and contribution-based role on real scenarios from SMAC.

| Scenario | Semantic Policy Diversity | Real Policy Diversity | Trajectory Diversity (overlap) | Contribution Diversity (max Q) |
|---|---|---|---|---|
| 4m_vs_5m | 1.5 | 9.1 | 0.47 | 0.13 |
| 3s_vs_5z | 2.7 | 18.7 | 0.21 | 0.09 |
| 4m_vs_4z | 3.3 | 19.3 | 0.31 | 0.06 |
| 4m_vs_3z | 3.8 | 12.1 | 0.35 | 0.25 |
| 1c1s1z_vs_1c1s3z | 8.7 | 22.0 | 0.40 | 0.03 |
| 1s1m1h1M_vs_3z | 2.4 | 13.2 | 0.41 | 0.61 |
| 1s1m1h1M_vs_4z | 2.7 | 15.8 | 0.25 | 0.75 |
| 1s1m1h1M_vs_5z | 6.2 | 22.5 | 0.18 | 0.82 |

Table 4: Different role diversities on different scenarios from SMAC. The minimum value of one column is labeled in green and the largest value is labeled in red. Detailed analysis can be found in Appx. E.

diversity and a variance term corresponds to the trajectory-based role diversity. Therefore, under the simple scenario in Sec. 4, we can find a clear relationship between different role diversity.

From the experiment perspective, the decomposition in (6) may not hold because of the more complicated settings. We have discussed this issue in the remark on Page 18. Here we collect all different role diversity data in Table. 4. We find scenarios like 3s_vs_5z have relatively small diversity in the policy-based role while the observation overlap of trajectory diversity is small. Scenarios like 4m_vs_5m have small policy-based role diversity while the observation overlap is large. Contribute-based role diversity can not be inferred from the policy diversity and trajectory diversity, and is more depending on the agents' behavior difference.

In conclusion, the relationship between different roles exists in MARL training theoretically, while this relationship is not so significant in the experimental perspective due to more complicated settings of the real MARL tasks. Yet the strong relation between different role diversity and the MARL training process still exists with no conflict with the conclusion of this paper.

## F   PARAMETER SHARING

Four different parameter sharing strategies are tested in our experiment including shared, no shared, partly shared, and selectively shared[8]. For partly shared, we only shared the GRU cell across different agents while keeping the embedding layer of the policy function model separated for each agent. For selectively shared strategy, we reproduce the grouping results following[8]. An illustration figure can be found in Fig. 9.

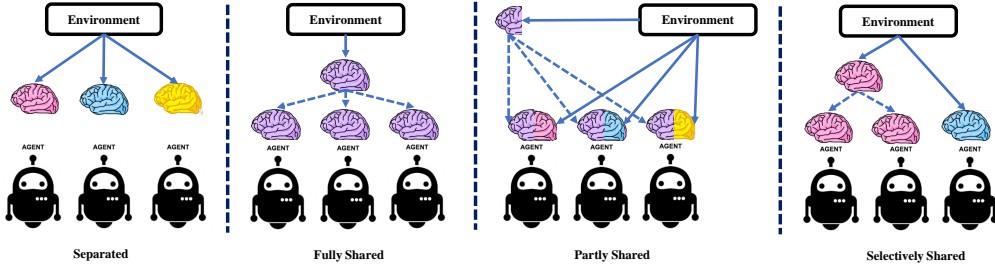

Figure 9: An overview of how knowledge sharing works with the MARL framework. Fully shared, partly shared, no shared (separated), and selectively shared[8] strategies are shown here. The same color indicates the same policy function part across different agents. Dash line represent only sharing no gradient backpropagation.

### F.1   SELECTIVELY SHARING THE PARAMETER

Here we provide the selective parameter sharing strategy result in Table. 5 as a supplement for Table. 1. The main purpose for doing so is to verify whether this method can serve as a general solution for parameter sharing strategy choosing issue. Selective parameter sharing strategy partitions the agents into the different groups automatically with an encoder-decoder model. However, the partition process is before the MARL training stage, which can not fully catch the policy difference. And according to the grouping result column in Table. 5, the selective parameter sharing strategy tends to divide agents by their initial attributes. This works well in scenarios like 1s1m1h1M_vs_4z and 1c1s1z_vs_1c1s3z but ignores the fact that the same type of agents may evolve to different functions during MARL training, which is the weakness of the selective parameter sharing strategy.

| Benchmark | Scenario | Role Diversity | Warm up | No shared | Shared | Selectively shared | Grouping results |
|---|---|---|---|---|---|---|---|
| | 4m_vs_5m | 1.5 / 9.1 | 6.5 | +3.6 / +4.4 | **+5.4 / +6.1** | **+5.4 / +6.1** | all shared |
| | 3s_vs_5z | 2.7 / 18.7 | 5.4 | +7.5 / +11.0 | **+8.2 / +11.8** | **+8.2 / +11.8** | all shared |
| | 2m | 3.1 / 12.2 | 6.0 | +9.2 / +11.1 | **+18.1 / +17.6** | **+18.1 / +17.6** | all shared |
| | 4m_vs_4z | 3.3 / 19.3 | 4.4 | **+ 8.8 / +12.7** | +5.4 / +8.4 | + 8.1 / +11.7 | 2m+2m |
| SMAC | 4m_vs_3z | 3.8 / 12.1 | 7.2 | +12.4 / +12.1 | +11.9 / +12.3 | **+12.6 / +12.2** | 2m+2m |
| | 3s_vs_4z | 5.2 / 32.5 | 4.8 | **+2.2 / +4.5** | +0.9 / +1.2 | +0.9 / +1.2 | all shared |
| | 1c1s1z_vs_1c1s3z | 8.7 / 22.0 | 11.8 | +4.1 / +6.1 | +2.7 / +5.4 | **+4.1 / +6.1** | 1c+1s+1z |
| | 1s1m1h1M_vs_3z | 2.4 / 13.2 | 16.2 | +3.4 / + 3.4 | **+3.4 / +3.6** | +3.4 / + 3.4 | 1s+1m+1h+1M |
| | 1s1m1h1M_vs_4z | 2.7 / 15.8 | 8.2 | **+7.8 / +11.6** | +5.3 / +10.0 | **+7.8 / +11.6** | 1s+1m+1h+1M |
| | 1s1m1h1M_vs_5z | 6.2 / 22.5 | 6.2 | **+6.4 / +9.1** | +3.7 / +6.1 | **+6.4 / +9.1** | 1s+1m+1h+1M |

Table 5: Model performance including selective parameter sharing as a supplement to Table. 1. The grouping result is provided in the last column.

## G   COMMUNICATION FRAMEWORK

The communication mechanism is important for MARL. The information shared can be location, action, and partial observation as showed in Fig. 10a. In many cases, communication is optional where the agent should learn when to communicate and how to ingest the information (dash line in Fig. 10b). In our experiment, we only consider the observation sharing method where the support set of policy functions contains both self partial observation and the aggregated observation information from other agents. The aggregated information is obtained by getting the mean value of other agents' observation and concatenate with the self partial observation. This means the support set of policy functions is now much similar to the global state.

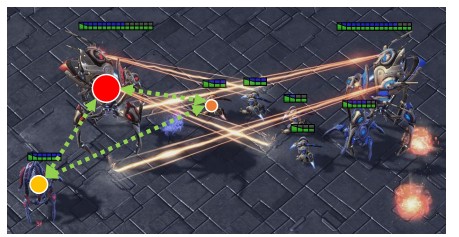
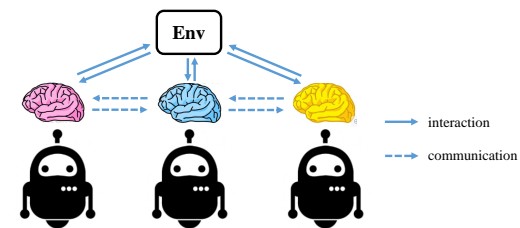

(a) Communicate with others: where I am, what I see and what I will do

(b) Communication is optional during optimization.

Figure 10: Communication works as a supplement part for MARL under the CTDE framework. (a) The sharing information can be current status or future policy, as the extra information for the decision making. (b) Learning when and how to communicate is critical to help policy learning.

## H CREDIT ASSIGNMENT

Credit assignment is the key part module for cooperative MARL, especially for the value decomposition-based method as it leverages the reward signal to each agent by approximate the $Q_{tot}$. Then the learned individual policies combine to form a joint policy interacting with the MAS.

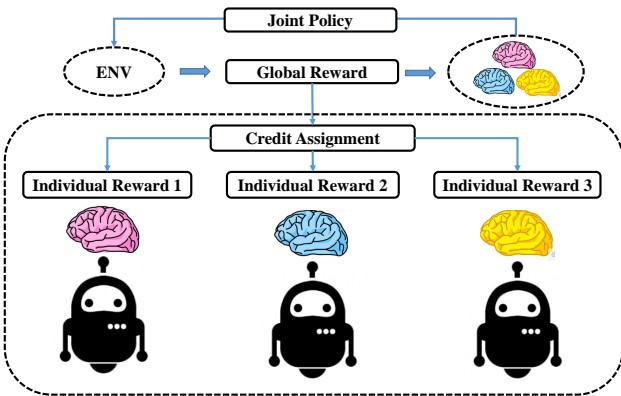

Figure 11: Credit assignment method focuses on assigning the proper individual reward from the total reward to update.

## I EXPERIMENT TABLE & CURVE

| Benchmark | Scenario | Sharing | IQL | IA2C | MADDPG | MAPPO | MAA2C | QMIX |
|---|---|---|---|---|---|---|---|---|
| MPE | Tag | no shared | 19.4 / 53.0 / 52.6 | 1.4 / 13.1 / 14.7 | 3.3 / 2.5 / 2.3 | 1.1 / 55.6 / 47.2 | 0.6 / 11.3 / 47.9 | 2.4 / 15.2 / 22.5 |
| | | shared ✓ | 16.8 / 50.3 / 47.9 | 1.0 / 16.6 / 27.5 | 3.1 / 5.9 / 32.8 | 1.4 / 40.0 / 45.9 | 0.8 / 42.1 / 60.9 | 2.9 / 23.3 / 36.0 |
| | Adversary | no shared | 15.8 / 16.3 / 16.7 | 17.1 / 19.7 / 19.9* | 16.8 / 19.0 / 16.0* | 18.8 / 20.1 / 20.8* | 15.3 / 19.6 / 20.4* | 13.3 / 16.1 / 16.5 |
| | | shared | 15.3 / 15.8 / 15.5 | 16.7 / 19.9 / 20.3* | 16.5 / 18.4 / 16.4* | 19.8 / 19.9 / 20.5* | 17.9 / 19.8 / 20.4* | 14.8 / 17.3 / 17.3 |
| | DoubleSpread-2 | no shared ✓ | 7.1 / 59.4 / 59.8 | 0.3 / 59.9 / 64.1 | 5.3 / 10.5 / 11.6* | 0.6 / 63.0 / 63.7 | 0.2 / 41.6 / 63.8 | 0.5 / 0.9 / 9.5 |
| | | shared | 4.3 / 5.4 / 8.0 | 0.2 / 7.9 / 11.1 | 5.3 / 10.5 / 11.6* | 3.1 / 25.6 / 56.5 | 0.3 / 10.1 / 19.2 | 0.6 / 4.3 / 6.0 |
| | DoubleSpread-4 | no shared ✓ | 32.2 / 144.3 / 212.2 | 12.3 / 436.4 / 480.8 | 1.1 / 1.2 / 1.3* | 47.0 / 261.4 / 261.1 | 4.7 / 343.6 / 390.5 | 3.2 / 3.2 / 2.9* |
| | | shared | 31.3 / 29.1 / 20.1 | 18.6 / 83.4 / 106.3 | 4.9 / 1.2 / 1.3* | 61.9 / 291.7 / 504.8 | 32.7 / 94.4 / 231.0 | 3.8 / 3.5 / 3.2* |
| SMAC | 4m_vs_5m | no shared ✓ | 4.8 / 7.6 / 8.1* | 6.4 / 6.6 / 6.7 | 2.5 / 2.4 / 1.1 | 6.9 / 7.1 / 7.2* | 6.6 / 7.0 / 7.1* | 7.0 / 9.9 / 10.9 |
| | | shared ✓ | 5.1 / 8.1 / 8.5* | 5.8 / 7.1 / 7.9 | 4.7 / 4.0 / 3.1 | 7.0 / 7.1 / 7.2* | 6.9 / 7.0 / 7.1* | 6.9 / 12.4 / 13.3 |
| | 3s_vs_5z | no shared | 4.6 / 5.1 / 7.9* | 4.2 / 4.3 / 4.4* | 2.8 / 4.1 / 4.5 | 4.3 / 6.0 / 6.1 | 4.1 / 4.4 / 5.3 | 4.6 / 13.5 / 17.0 |
| | | shared ✓ | 4.3 / 5.3 / 7.8* | 4.1 / 4.4 / 4.4* | 3.3 / 4.5 / 5.1 | 5.7 / 6.9 / 6.5* | 4.1 / 5.8 / 6.0 | 4.2 / 12.9 / 20.0 |
| | 1c1s1z_vs_1c1s3z | no shared ✓ | 10.8 / 12.3 / 12.2 | 11.0 / 11.4 / 11.5 | 9.5 / 13.4 / 13.5* | 11.1 / 12.9 / 13.5 | 10.2 / 11.6 / 12.2 | 12.9 / 17.8 / 19.4 |
| | | shared | 9.8 / 11.2 / 11.9 | 9.0 / 9.2 / 9.4 | 9.7 / 12.8 / 13.4 | 10.7 / 12.2 / 12.6 | 9.7 / 10.6 / 11.0 | 12.5 / 15.8 / 18.4 |
| | 4m_vs_4z | no shared ✓ | 3.3 / 3.2 / 3.7 | 2.6 / 4.0 / 5.4 | 2.4 / 2.0 / 3.0 | 4.3 / 4.5 / 4.5* | 4.0 / 4.6 / 4.5* | 4.3 / 18.3 / 18.8 |
| | | shared | 2.6 / 3.2 / 3.2 | 3.7 / 3.9 / 3.9 | 3.2 / 2.9 / 1.4 | 4.1 / 4.2 / 4.3* | 3.8 / 4.1 / 4.2* | 4.3 / 14.8 / 16.5 |

Table 6: Policy-based role diversity influence the performance of different parameter sharing strategies on the MPE [28] and SMAC [39] benchmarks.

| scenario | obs overlap | baseline | communication | scenario | obs overlap | baseline | communication |
|---|---|---|---|---|---|---|---|
| 4m_vs_5m | 0.47 | 6.5 / 10.1 / 10.9 | 6.6 / 11.5 / 11.4 +1.4 +0.5 | 4m_vs_3z | 0.37 | 7.5 / 19.6 / 19.3 | 7.7 / 19.6 / 19.4 0.0 +0.1 |
| 1c1s1z_vs_1c1s3z | 0.40 | 12.3 / 15.9 / 17.9 | 12.4 / 15.7 / 18.1 -0.2 +0.2 | 3s_vs_5z | 0.21 | 5.4 / 12.9 / 16.4 | 5.4 / 12.4 / 15.5 -0.5 -0.9 |
| 4m_vs_4z | 0.32 | 4.3 / 13.2 / 17.1 | 4.1 / 15.9 / 18.3 +2.7 +1.2 | 1s1m1h1M_vs_4z | 0.25 | 8.4 / 16.0 / 19.8 | 7.9 / 13.2 / 19.0 -2.8 -0.8 |

Table 7: Model performance with and without communication. The performance is recorded in the 'warmup performance / half steps performance / final performance' pattern. Detailed analysis can be found in Sec. 5.2.

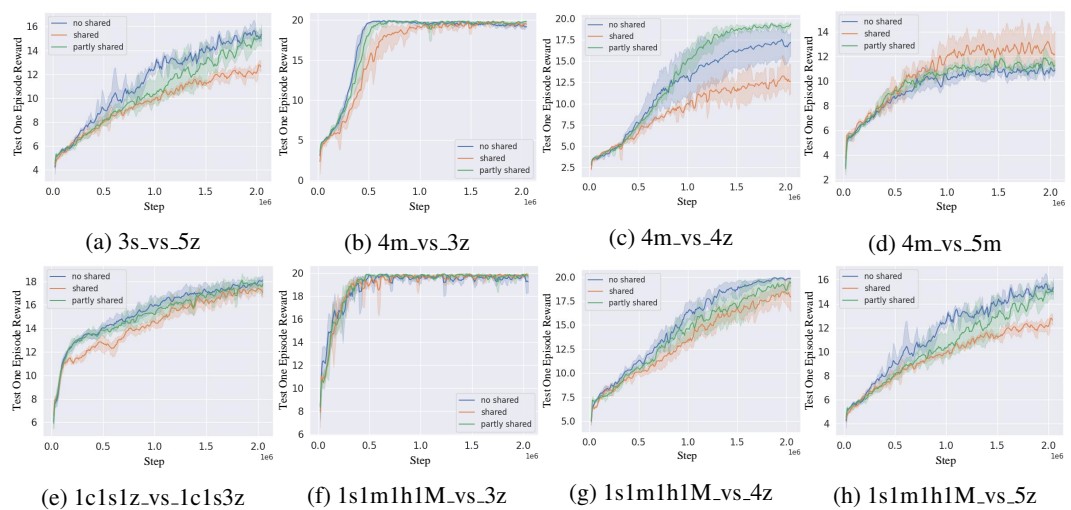

Figure 12: Policy learning curve with different parameter sharing strategies.

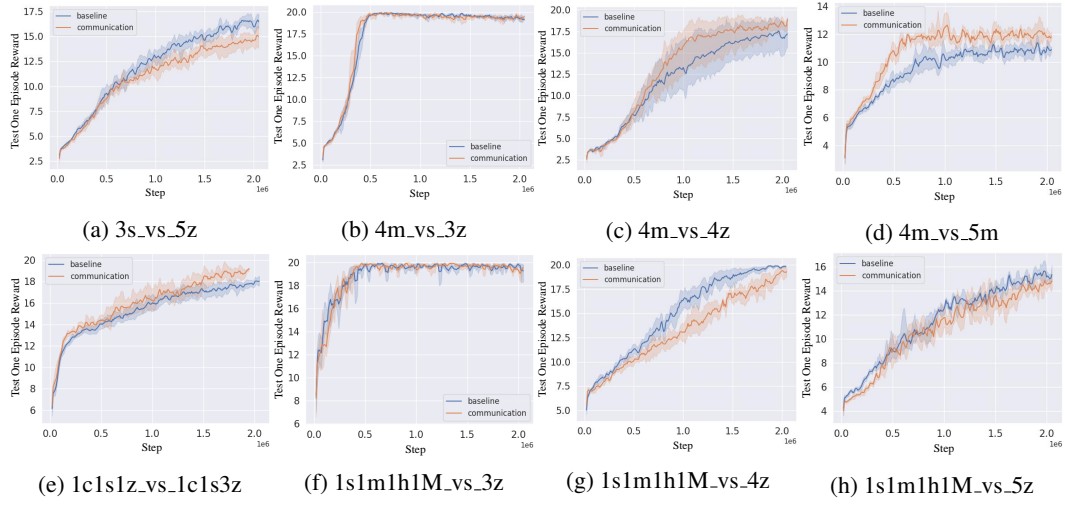

Figure 13: Policy learning curve with and without communication on different scenarios.

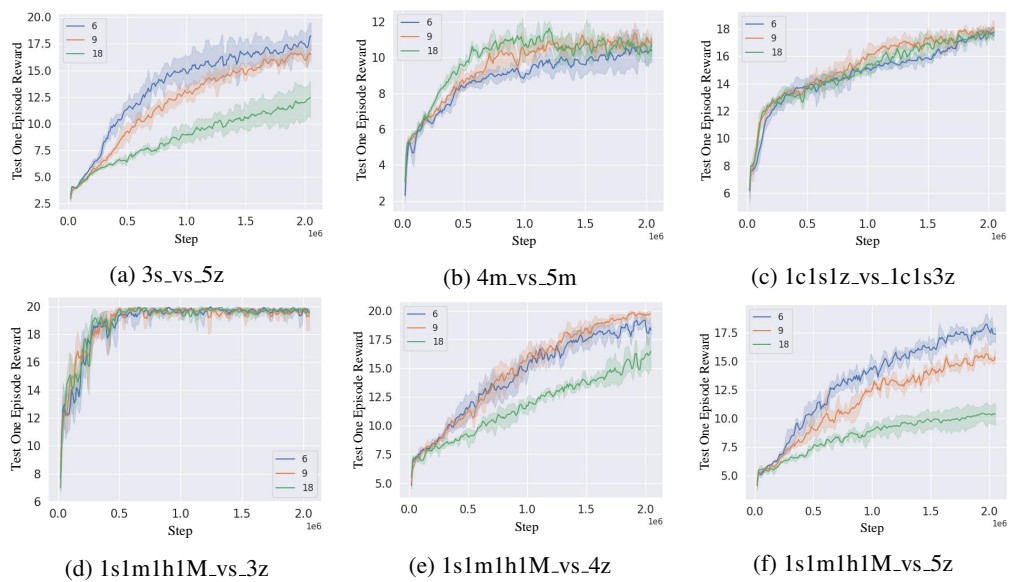

Figure 14: Policy learning curve with different vision scope (6-9-18).

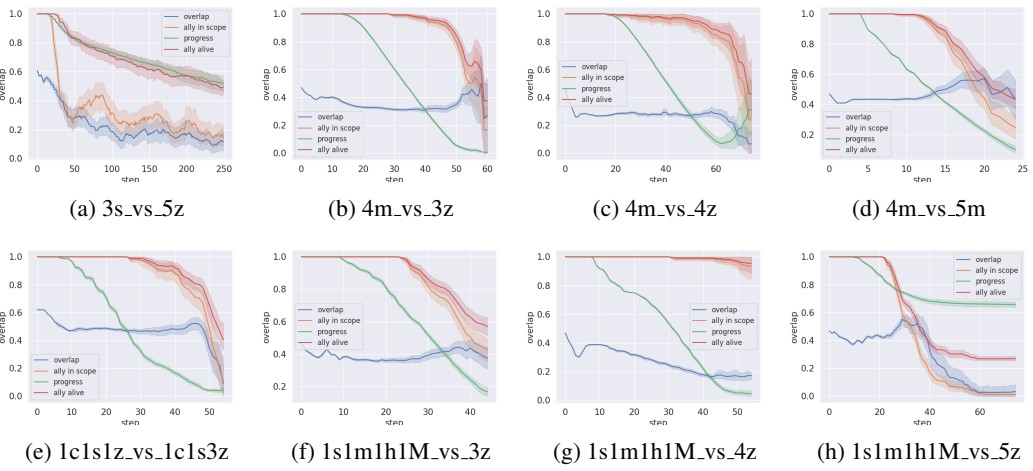

Figure 15: Observation overlap curve of one episode game on different battle scenarios. The policy is trained using VDN[48] and no parameter sharing. We also provide the curve of *game progress*(equals to the enemy health), *ally in scope* and *ally alive*. All values are normalized from 0 to 1.

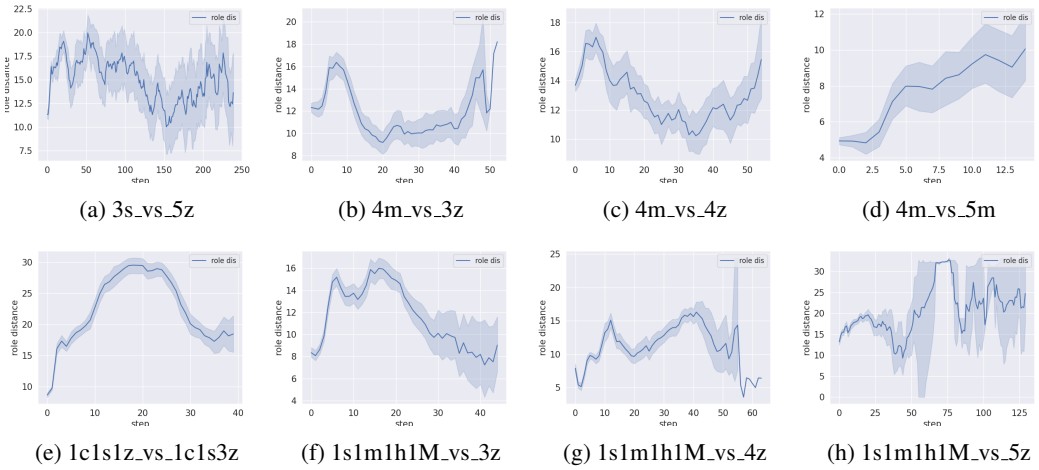

Figure 16: Policy based role diversity(real) in one episode.

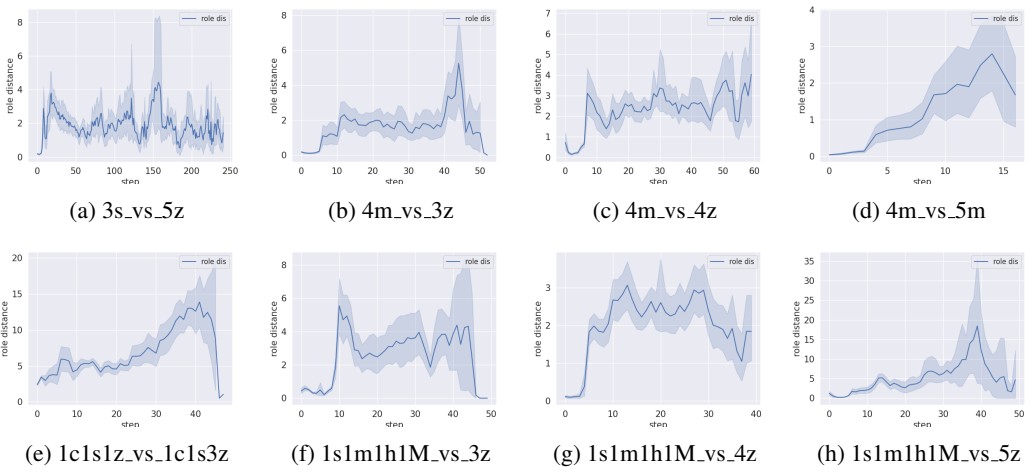

Figure 17: Policy based role diversity(semantic) in one episode.

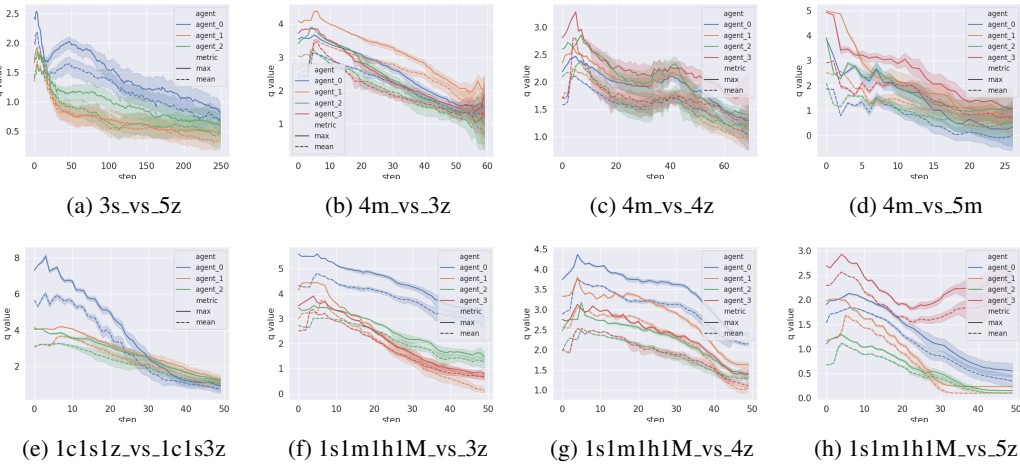

Figure 18: $Q$ value curve in one episode on different scenarios.

