# OpenReview forum: "Role Diversity Matters: A Study of Cooperative Training Strategies for Multi-Agent RL"
_ICLR.cc/2022/Conference — ICLR 2022 Submitted_

### Official Review · Reviewer_mf45 · 2021-11-01

**Correctness:** 3
**Technical Novelty And Significance:** 2
**Empirical Novelty And Significance:** 3
**Recommendation:** 5
**Confidence:** 3

**Main Review:**


-- Positives --
- Role diversity is important in multi-agent environments. Being able to quantify it might allow us to better understand how to improve multi-agent reinforcement learning. I also agree that such a metric can help us understand why some MARL algorithms perform so differently across a range of tasks.

-- Negatives --

1. - (3.1 - Policy-based role): The frequency of actions does not seem to be a reliable metric for determining the role of the agent for the following reasons: i) there is no guarantee that actions do the same thing (e.g. action '3' can mean different things for a "marine" and a "zergling" in the game of SMAC), ii) not considering the _order_ that the actions are selected loses important information (e.g. in a maze, going left and then right is significantly different to going right and then left). Therefore I believe focusing on the trajectory as a whole makes much more sense and includes the frequency of the actions of this paragraph.

2. - (3.2 - trajectory-based role): The "overlap percentage" of the observations is not general enough and does not seem straightforward to calculate in all environments. The authors present an example for the SMAC task in the appendix, but SMAC happens to have a circular observation radius which is then easy to calculate. There are many environments where the observations cannot be easily compared (e.g. a simple image, or even features that describe the state of the environment without a specific physical representation). In addition, fully-observable environments are not uncommon and, in my understanding, the rule breaks down in that case (the distance between observations is always 0).

3. - In general, all those metrics seems to require that training has already been completed. The contribution-based role specifically requires (trained, because I assume untrained aren't useful) q-values. As such, I cannot see how these metrics can help with the training of MARL agents, since they require those trained policies, to begin with. The same should apply to the policy-based rule. In my understanding, this is not in line with the motivations and promises of this work, that a diversity metric can help with a better training strategy and performance in MARL.

Other/fixable:

4. - (5.1 parameter sharing): I would expect different roles to require different parameters and roles that are similar to benefit from shared parameters (which is also shown in "...selective parameter sharing..."[7]). You could consider a similar parameter sharing scheme on top of the "no/partly/fully" shared parameters.

- While the writing is clear, and conveys the author's intentions, it could certainly be further polished. There are some sentences that are vague or that could benefit from rewriting (e.g. first page, "In other word, even adopting the ... performance", but there are other sentences as well). The document could benefit from some proofreading.

- The bibliography is untidy and needs to be further polished. Many of the references inaccurately state "arxiv" while they have been published in conferences.
- Using citations as a noun is discouraged (e.g. "In [52], the role]...")
- (page 5) acroess -> across

**Summary Of The Paper:**

The authors propose a role diversity metric to quantify the difference between agents in multi-agent reinforcement learning. The metric is used to inform the use of communication, parameter sharing, and other MARL algorithmic decisions.

**Summary Of The Review:**

For the stated reasons, the proposed diversity metric does not manage to convince me of its potential usefulness, although I still believe that the research direction is promising.

---

> ### Author Response · Authors · 2021-11-19
> **Response to Reviewer mf45 Part 2**
>
> > Q2. (3.2 - Trajectory-based role): The "overlap percentage" of the observations is not general enough and does not seem straightforward to calculate in all environments...
>
> **Ans:**
>
> We agree that the observed overlap calculation method we proposed in our paper is easy to get in games like SMAC and MPE, but it can be applied to real scenarios like embodied vision robots (which take a single image as input) with a revised trajectory distance $d_t$. We claim that even in complicated real-world tasks, the ‘overlap’ concept can still serve as a good substitution for computing trajectory-based role diversity. We proposed three methods for calculating the trajectory-based role distance in popular multi-agent VLN tasks based on our ‘overlap’ concept and are compatible with Eq.4 to get the trajectory diversity. The illustration of this newly added part can be found in Appx. C, where we discuss both semantic overlap and image feature overlap based on VLN tasks with a single image as input. We hope this may be a good attempt to extend our raw method from game scenarios to real scenarios.
>
> We also notice that currently, multi-agent systems designed and proposed in the MARL community have quite diverged and the setting varies. We are not saying that observation overlap in SMAC is the only way to measure the $d_t$, but want to emphasize that it is a good way to avoid directly measuring the distance between two trajectories, which can be troublesome in many cases. In addition, fully-observable environments are not considered in this paper as most MAS are under POMDP settings.
>
> > Q3. In general, all those metrics seem to require that training has already been completed. The contribution-based role specifically requires (trained, because I assume untrained aren't useful) q-values. As such, I cannot see how these metrics can help with the training of MARL agents, since they require those trained policies, to begin with...
>
> **Ans:**
>
> In this paper, we focus on finding the reason why the performance of different MARL training strategies varies in different scenarios. That is our key point and main contribution. We want to better understand the task difference and give proper metrics to describe this difference, and finally, find the connection between this description and the model performance from both theoretical analysis and experiment results based on role diversity. We are not developing metrics to improve one MARL algorithm’s performance, like modifying the training pipeline or proposing a new regularization term. We are discussing all kinds of MARL algorithms combined with different training strategies(parameter sharing, communication, and credit assignment), and explaining what may bring improvement or degradation to the algorithm performance. For instance, if the contribution-based role diversity of a scenario is large, we should be careful using learnable mixing network-based algorithms like QMIX, which may cause significant performance degradation. Only by accurately describing different multi-agent cooperation tasks can we get an understanding of how to choose the proper training strategies, and with no need to waste time on testing a bunch of other algorithms. And it is especially beneficial for subsequent adjustment to the developed algorithm to avoid the possible bottlenecks and pursue a higher performance. This is what we want to deliver in this paper and the major motivation of this study.
>
> To do this, we need a pretrained policy to describe a scenario. As we mentioned in Sec. 6, in this study, all the role diversity values come from VDN and separated training with no communication because we require the baseline method in our study to be robust and training-efficient. Therefore, the diversity calculation process may cost extra training time. But compared to the time saving from improper MARL algorithms selection and training, it is worth first getting the role diversity description for the current task and then deciding which algorithm to use or what adjustment we should make based on this description.
>
> > Q4. (5.1 parameter sharing): I would expect different roles to require different parameters and roles that are similar to benefit from shared parameters (which is also shown in "...selective parameter sharing..."[7]). You could consider a similar parameter sharing scheme on top of the "no/partly/fully" shared parameters.
>
>
> **Ans:** Thank you for the suggestion. Due to the page limitation, we provide both related experiment results and discussions in Appx. F. As a conclusion, selective parameter sharing is a good choice but still has shortcomings.
>
> > Q5. While the writing is clear and conveys the author's intentions, it could certainly be further polished. ... ... (page 5) acroess -> across
>
> **Ans:** Thank you for the comments. In the revision, we have corrected the typo errors,  proofread the paper, and polished the bibliography.

---

> ### Author Response · Authors · 2021-11-19
> **Response to Reviewer mf45 Part 1**
>
> We thank you for your time and valuable feedback. We answer your questions here.
>
>
> > Q1(i). (3.1 - Policy-based role): The frequency of actions does not seem to be a reliable metric for determining the role of the agent for the following reasons: there is no guarantee that actions do the same thing (e.g. action '3' can mean different things for a "marine" and a "zergling" in the game of SMAC).
>
> **Ans:**
>
> The experiment conducted in this paper is based on SMAC and MPE, where the actions are aligned (the action space is fixed and actions with the same index have the same meaning like move or attack). Despite this, actions that have the same meaning may have different behaviors or impacts on the environment. For instance, attack actions of zerglings (melee) and marines (range) are quite different (as mentioned by the reviewer). From this perspective, it looks like our definition by merely counting the action frequency cannot catch this difference.
>
> However, because of this difference, the action frequency can be so different: the various styles/impact of attacking behavior force the agent to learn the balance between the frequency of attack actions and move actions. As a result, the policy of zerglings intends to attack only or move only in a period (which is known as the attack and escape strategy). While marines intend to alternately use move action and attack action (move out of enemy scope, return to attack, like kiting the enemy). Therefore, we claim that action frequency-based policy diversity can catch more information behind the action frequency itself. It reflects the agent’s other attributes like attack behavior, move speed, and character (brave, timid).
>
> In addition, if we treat the whole multi-agent system as a black box, even actions with the same index have different physical meanings, the claim that policy-based role is strongly related to parameter sharing strategy still holds. From the MARL training perspective, there is no need for the model to be aware of the action meaning. For instance, if the optimal individual policies are similar, the model is good enough when it learns to output the same action index for different agents. It is not necessary to consider exactly the impact or behavior of the actions.
>
>
>
> > Q1(ii). (3.1 - Policy-based role): The frequency of actions does not seem to be a reliable metric for determining the role of the agent for the following reasons: not considering the order that the actions are selected loses important information (e.g. in a maze, going left and then right is significantly different to going right and then left).
>
> **Ans:**
>
> We agree with the reviewer that the order information is very important. And a better metric to include this information will be the trajectory-based role as the reviewer mentioned, which automatically contains the action order information. However, we claim that the policy-based role is still a necessary metric and cannot be replaced by the trajectory-based role. In Table 1 and Table 2, we find some scenarios that have relatively small policy-based role diversity but large trajectory-based role diversity, e.g. 3s_vs_5z. This indicates that these two metrics can be very different. We collected metric data of all three role diversities from different scenarios in Table 4. And discussions are held in Appx. E to study the connections between them.
>
> Besides, to explain the connection between different role diversities and their relations to the different MARL training strategies more clearly, we have revised Sec. 4. Please refer to the last paragraph of Sec. 4 as a summary of these relations.

---

> > ### Comment · Reviewer_mf45 · 2021-11-29
> > **Reply to author clarifications**
> >
> > Part I
> > I am still not convinced that the policy-based metric can generalise beyond the SMAC game. I agree that the experiments suggest that a policy-based role diversity can be small and the trajectory-based role diversity large: but this is one of my original points, that the policy-based role calculation is not able to sufficiently capture the diversity between policies (because it ignores order information) and might mistake two different roles as similar (small diversity as mentioned in your reply while the trajectory role diversity is actually large).
> >
> > Part II
> > I agree that an "overlap" often exists in real scenarios and can be a useful metric. The embodied vision robots is a convincing example. That said, I still do not understand how this can generalise to other environments, which I believe is necessary for motivating this particular metric.
> >
> > I am now less concerned for the motivation of this paper. Finding the reason of the different training performances is worthwhile in my opinion . However, I believe that the paper can better explain this intention to the reader (e.g. section 6 implies that we can use this information to find better strategies).
> >
> > With these clarifications and additional experiments, I believe the paper has improved during the rebuttal period. But, some of my points have not been completely answered and thus I have decided not to increase my score. I would not be upset however if the paper was accepted.

---

> > > ### Author Response · Authors · 2021-11-29
> > > **Reply to Reviewer mf45**
> > >
> > > Thank you very much for your response.
> > >
> > > We believe that our work presents clear experimental and theoretical evidence to show the utility of our proposed metrics in most POMDP MAS. And we want to emphasize that there is no need for one role type (like policy-based) to cover all the information (like action order) because other role types (like trajectory-based) will catch it.
> > >
> > > We accept that the metrics may not be general and perfect due to various task settings in MARL. We will study the generalization properties of the diversity metrics in future work. Thank you for the constructive comments.

---

### Official Review · Reviewer_SqGu · 2021-11-02

**Correctness:** 3
**Technical Novelty And Significance:** 3
**Empirical Novelty And Significance:** 4
**Recommendation:** 6
**Confidence:** 4

**Main Review:**

The reviewer finds the paper is rather timely and interesting. With so many algorithms being proposed, how to choose one is quite challenging sometimes. This paper proposes to use role diversity to describe the MARL tasks and the experiments strongly show the relationship between algorithm choosing and role diversity. However, the reviewer also thinks the paper needs to be improved.

(1) The paper does not provide enough discussion or experiment about the connections and differences between these three role diversity definitions. They might describe different diversity aspects. But there also might be an underlying diversity definition behind these three definitions. Another question is that is one role diversity definition (such as policy-based) for one algorithm design (such as parameter sharing)? Are there any connections?

(2) Theoretical analysis is not clear enough. The reviewer appreciates that this paper provides formal analysis, but the reviewer is confused about how the empirical role diversity definitions are linked to theoretical analysis. Specifically, more explanations about Eq. (5-6) are expected.

(3) The definitions of trajectory-based and contribution-based diversities should be explained in Eq. (3-4). For example, what are the definitions of $d_T$ in these two equations? How to compute them?

(4) The discussion of how to obtain an accurate measurement of the role diversity before training in Sec. 6 could be moved to Sec. 3.

(5)  What do the figures in Fig.2 (b)(c) want to say?

(6) Some minor errors. It should be $T$ instead of $t_0$ when describing the notations of Eq. (2).

**Summary Of The Paper:**

To address the problem of algorithm choosing in different MARL tasks, this paper proposes to use role diversity as a metric to describe MARL tasks. For choosing parameter sharing, communication mechanism, and credit assignment strategy, this paper defines three role diversity metrics, i.e. policy-based, trajectory-based, and contribution-based, respectively. They also find that the error bound in MARL can be decomposed into three parts that have a strong relation to the role diversity. To evaluate the proposed method, they further conduct some experiments on MPE and SMAC environments.

**Summary Of The Review:**

This paper is timely and interesting, but there are still some concerns, especially the the first one in the Main Review. The reviewer thinks this paper should be further improved.

---

> ### Author Response · Authors · 2021-11-19
> **Response to Reviewer SqGu**
>
> We thank you for your time and insightful questions. We are encouraged by positive feedback. Here we address the concerns mentioned in the Main Review one by one.
>
> > Q1(i). The paper does not provide enough discussion or experiment about the connections and differences between these three role diversity definitions. They might describe different diversity aspects. But there also might be an underlying diversity definition behind these three definitions.
>
> **Ans:** To provide a more comprehensive description of these relations, we revise the paper and analysis both theoretically and experimentally.
>
> From the theoretical perspective, the contribution-based role diversity is a compound description of role diversity. It corresponds to the variance term in Eq.(5). For the parameter sharing case, we decompose this variance into a sum of two terms: a bias term corresponds to the policy-based role diversity and a variance term corresponds to the trajectory-based role diversity. Therefore, under the simple scenario in Sec. 4, we can find a clear relationship between different role diversity.
>
> From the experiment perspective, the decomposition in (6) may not hold because of the more complicated settings. We have discussed this issue in the remark on Page 18.  Here we collect all different role diversity data in table 4 and add additional illustrations in Appx.E to further discuss this.
>
> In conclusion, the relationship between different roles exists in MARL training theoretically, while this relationship is not so significant in the experimental perspective due to more complicated settings of the real MARL tasks. Yet the strong relation between different role diversity and the MARL training process still exists with no conflict with the conclusion of this paper.
>
> > Q1.(ii) The relationship between algorithm design (such as parameter sharing) and different role diversity (such as policy-based) is not clearly explained.
>
> > Q2. The theoretical analysis is not clear enough to explain the relation between empirical definition and theoretical analysis.
>
> **Ans:**
> We would like to answer these two questions together.
>
> Due to the page limitation, we put most of the theoretical analysis in Appx. A and only leave the main conclusion on Sec.4. To help better understand the link between empirical role diversity definitions and theoretical analysis, we have revised the last paragraph of Sec. 4 as a summary of the empirical definition-theoretical analysis connection to help understand. We also provide explanations for Eq.(5-6) on page 6. More detailed proof and explanation can be found in Appx.B.  In conclusion, the connection is clear between the empirical definition of role diversity and theoretical analysis on MARL and is proved to have a significant impact on the model performance in the experiment part.
>
>
> > Q3. The definitions of trajectory-based and contribution-based diversities should be explained in Eq. (3-4). For example, what are the definitions of these two equations? How to compute them?
>
>
> **Ans:**
> To address this concern, we give examples in Appx.C, covering both game (SMAC) and real-world tasks (multi-agent VLN) with three different methods including radius-based overlap, semantic-based overlap, and image-based observation overlap. Contribution-based role distance is defined as the absolute value difference of Q value or state value between two agents in Sec.3.3. We slightly modify Eq.(4) to normalize the total diversity to (0, 1) which is suggested by Reviewer 97k6.
>
> > Q4. The discussion of how to obtain an accurate measurement of the role diversity before training in Sec. 6 could be moved to Sec. 3.
>
>
> **Ans:**
> Thank you for the suggestion. In the revision, we have added an explanation of how we get all the diversity data from different scenarios at the end of Sec. 3.
>
> > Q5. What do the figures in Fig.2 (b)(c) want to say?
>
> **Ans:**
> We provide Fig.2 (b)(c) to show the different role diversity in different scenarios on both trajectory-based (b) and contribution-based (c). More curves can be found at the end of Appx.H from rich scenarios. We pick the most representative scenario to show in Fig.2 (b)(c) where 3s_vs_5z is the typical map with large trajectory-based role diversity while 4m_vs_5m is the opposite, and 1s1m1h1M_vs_5z is the typical map with large contribution-based role diversity while 4m_vs_3z is the opposite.
>
> > Q6. Some minor errors. It should be $T$ instead of $t_0$ when describing the notations of Eq. (2).
>
> **Ans:** Thank you for the comments. We have corrected these errors.

---

### Official Review · Reviewer_L6y7 · 2021-11-02

**Correctness:** 2
**Technical Novelty And Significance:** 2
**Empirical Novelty And Significance:** 2
**Recommendation:** 5
**Confidence:** 2

**Main Review:**


Strengths:

* Interesting question of how to define roles in multiagent systems and apply them to analyzing policy behavior.

Weaknesses:

* The paper is not well written. The paper does not do a good job at motivating why we should care about analyzing multiagent systems from the perspective of roles, or the alternative role definitions they propose.

I would recommend the authors re-frame the paper narrative and improve the writing quality; I’d made suggestions under “Summary of review”.

* The paper does not do a convincing job at showing that the roles are core to “cooperative training strategies for multiagent RL”.

Assuming we do think roles are an important way of understanding multiagent behavior, the role perspective only shines light on agent patterns, but don’t necessarily cause improved cooperative strategies. A convincing suite of experiments that would need to be conducted would be if agents were explicitly optimized for those particular metrics with minimal impact to the nominal performance. That would be the right test to show that the decomposition does in fact play an important role in policy optimization, which is what the authors claim in their abstract.

* The theoretical analysis is not well-motivated and it’s not clear whether it provides any operationalizable insights. It adds more confusion to their role diversity framework.

For example, the authors try to show that their policy-based and contribution-based role can be tied to an approximation error between the optimal joint and independent policies. What’s confusing is that their analysis suggests that agents should minimize these role metrics in order to perform closer to the optimal joint distribution. This conclusion doesn’t make sense since the work’s experimental setup and narrative is that these metrics are important in improving the policy quality.



**Summary Of The Paper:**

This work looks at defining role diversity as a means for analyzing multi agent dynamics. It proposes three different perspectives on analyzing roles in multiagent systems. One is from a policy perspective (based on KL divergences), an trajectory perspective (how much do the agents overlap in what they observe), and a team-contribution perspective (weighing the effects of the agent to the team reward).

**Summary Of The Review:**


--- motivation ---
I’m not convinced that better understanding roles will lead to improvements in multiagent coordination. In fact, this is not really the investigation the paper conducts: The paper demonstrates there are certain correlations between their role definitions and coordination performance, but not that this understanding results in coordination improvements. What concrete challenges currently exist in multiagent optimization --- e.g. what do the authors mean by `roles can avoid the issue of “policy degradation”` in the abstract? --- and how can these challenges be addressed by understanding how groups/teams can be factorized into roles?

I’m confused why the authors break down MARL cooperative algorithms by “parameter sharing”, “communication” and “credit assignment”. These are components that don’t seem directly comparable to each other, and it’s not clear how roles are related to the components. Parameter sharing is a technical means to learning policies, communication is an assumption by which we allow additional agent interaction / information passing, and credit assignment is a general challenge when optimizing in long-horizon settings for reinforcement learning.

The authors directly propose their three different role definitions in the second to last paragraph of the introduction, but these alternative definitions need more context and motivation. It’s not clear what challenges they address and why these definitions are ones we should use for characterizing multiagent behavior. Additionally, without context or grounding in prior work, it’s unclear what “policy-based,...” roles should even refer to.


--- method ---

How does this proposal show that agents take on the same role? If I’m finding a good position just like all other agents, shouldn’t this indicate that we have the same role? This doesn’t seem reflected in this metric. The policy-based role also seems to have a lot of overlap with the trajectory-based role --- in some sense, the trajectory based role feels quite redundant: If I can infer what role the agent has, I should be able to have a good generative model on what the agents’ most likely trajectories should look like.

In Equation 1, a sum over the policy conditioned on an action is an unnormalized probability distribution. It’s not clear to me how this is being used in the KL metric.

In Equation 2, the normalization should happen over A^2, and not A.

It seems weird that the trajectory-based role is defined as the observation overlap. Observation overlap seems to be a measurement of … pairwise observability (i.e. how much information do I have about what you see, and you about what I see?), and less about roles.

I disagree with the authors on their interpretation of Q values on “contribution-based roles”: in the setting where agents are performing individual policy optimization, I don’t see how Q values [optimized to reflect the agents’ expected return, and not a ratio between the outcomes of an agent’s actions and all other teammates] should not be interpreted as the agent’s contribution to the team. If there’s a misunderstanding or some related work that asserts this, the authors should elaborate.

--- theoretical analysis ---

The notation is not self-contained in this section, e.g. “Please refer to Sec. B.1 for the detailed definitions”.

The second term in the error is the irreducible error after we find a best fit for relating the optimal joint Q values and the concatenation of independent Q values. How do we go from this line to line (5)? On a related note, how are we going from line (5) to (6)?

“The first term on the RHS of (5)...” I don’t see how this difference is related to the contribution based role which is based on an agent’s Q values. How can this be related to the linear weights?

“The first term on the RHS of (6)...” Minimizing the bias results in minimizing the policy-based metric, which intuitively leads to agents _not_ aligning their actions. Does it make sense to think about this minimization as reducing the error we incur when approximating a joint policy with independent policies? This seems contradictory to what the authors want to propose in their methods section. The bias term should also normalized over the number of samples too.


--- claims ---
“These definitions are intuitive and cannot accurately describe the role difference.” Why not? What is insufficient with their definitions? It would be nice if these definitions could also be used in the experiments to compare and contrast the success/failure modes.

“As common sense would indicate, actions taken at the same time step can indicate different roles” - Do actions do this? This is arguable. Actions can be considered coordinated or uncoordinated, joint vs. independent. They can indicate different or shared roles.

--- figure ---
Figure 1 is quite confusing for the following reasons
> What is the blue vs. green color schema supposed to refer to?
> I don’t understand what MARL alg 1 and 2 should represent. I thought the agents are trained using the same algorithm?
> Related to my comment on “why the authors break down MARL cooperative algorithms by ... ”, I don’t understand why these axes are being used as input parameters for the algorithm in Fig 1a, and then as outputs from Role Diversity.
If the point is that role diversity should inform our assumptions in developing algorithms, then I’d recommend rephrasing the figure caption, omitting the performance graphs, and improving the figure pipeline. The figure doesn’t elucidate the authors claim that “role diversity [avoids] possible bottleneck of a MARL algorithm … ” What is this bottleneck?

Figure 2 onwards refer to the scenarios with hard-to-parse names, eg. “1s1m1h1M_vs_5z”. These figure names should be updated for clarity.



--- writing nits ---
Introduction
> “lack the study of the question of why” - too wordy and awk.
> “In other word” → “In ”
> “Current researches are more focusing”
> “The reason maybe:” - informal.

Related works
> “adopt q-learning” - note capitalization
> “In this study, we contended that” - note tense

---

> ### Author Response · Authors · 2021-11-19
> **Response to Reviewer L6y7 Part 4**
>
> > Q3(v). “The first term on the RHS of (6)...” Minimizing the bias results in minimizing the policy-based metric, which intuitively leads to agents not aligning their actions. Does it make sense to think about this minimization as reducing the error we incur when approximating a joint policy with independent policies? This seems contradictory to what the authors want to propose in their methods Sec. The bias term should also be normalized over the number of samples too.
>
> **Ans:**
> The bias term in Eq.(6) is caused by the assumption that all agents share the same Q-function. When the Policy-based Role diversity is significant, this bias term will be large and cannot be optimized by the optimization procedure. This is the price of using parameter sharing. In addition, both $Q^*_i$ and $\bar{Q}^*$ are the optimal solutions under the population level and $\mu$ is a given distribution. So the bias term is independent of the training data.
>
> --- claims ---
>
> > Q4(i). “These definitions are intuitive and cannot accurately describe the role difference.” ...It would be nice if these definitions could also be used in the experiments to compare and contrast the success/failure modes.
>
> **Ans:**
> We have introduced the role definition in RODE[1] and SEPS[2] in the first paragraph of Sec.3.
>
> In RODE, the role is defined as the higher-level option in the hierarchical RL framework. They use unsupervised clustering on the action embedding before the policy output. The number of clusters corresponds to the role number. However, this role definition only focuses on agent action on one timestep, without considering the role in a period. And other important information like trajectory distance and attribute differences are not included, which makes this role definition not so comprehensive.
>
> In work SEPS, the role is defined as the environmental impact similarity of a random policy. This role definition is a good choice to help decide the parameter sharing strategy without human prior. However, it has significant shortcomings. We reproduce the model performance using SEPS based on SMAC in Table 5. Further analysis can also be found in Appx.F.1.
>
> > Q4(ii). “As common sense would indicate, ... They can indicate different or shared roles.
>
> **Ans:**  We do not agree with this and propose a new definition of the role concept.
>
>
>
> --- figure ---
>
> > Q5. What is the blue vs. green color schema supposed to refer to?...What is this bottleneck?
>
> **Ans:**
> We choose parameter sharing strategy, communication, and credit assignment because these three components are mostly discussed in MARL. They significantly impact the model performance and are the main topics in recent cooperative MARL research. These axes are not used as input parameters but are the component for the final algorithms. For instance, one algorithm can consist of a fully parameter sharing strategy, no communication, and QMIX as the credit assignment module. In (a), without the role diversity, the performance of different combinations of these training strategies varies on different scenarios and it is hard to decide which one to choose. With the role diversity description in (b), we now first measure the role diversity of each scenario. Then we decide which training strategies are more suitable for the current task. And finally we combine them to form the best algorithm for Task n.
>
> The bottleneck comes from adopting improper training strategies in the final algorithm. For instance, using parameter sharing strategy in scenarios with large policy-based role diversity or using credit assignment module with a learnable mixing network in scenarios with large contribution-based role diversity can be the bottleneck for final performance even if other training aspects are suitable for current tasks. This is also known as the barrel effect.
>
> --- writing nits ---
>
> > Q6 “lack the study of the question of why” ...- note tense
>
> **Ans:**
> Thank you for the comments. We have corrected these errors in the revision.
>
> We sincerely hope our response can clarify your concerns and convince you to increase the score. Any further discussion is welcome.
>
> [1] RODE: Learning Roles to Decompose Multi-Agent Tasks
>
> [2] Scaling Multi-Agent Reinforcement Learning with Selective Parameter Sharing

---

> ### Author Response · Authors · 2021-11-19
> **Response to Reviewer L6y7 Part 3**
>
> --- theoretical analysis ---
>
> > Q3(i). The notation is not self-contained in this Sec, e.g. “Please refer to Sec. B.1 for the detailed definitions”.
>
> **Ans:**
> In Sec. 4, we have explained all the notations involved in the theoretical results. But the definition of some symbols is very complex and requires a lot of space to give a rigorous mathematical formulation, e.g. the concentration coefficient $\phi_{\mu,\nu}$ and the cover number $N_0$. So some details of the definitions are postponed into Appx.B.1. We have gone through this Sec. and made sure that the logic of the theoretical results here is complete.
>
> > Q3(ii). The second term in the error is the irreducible error after we find the best fit for relating the optimal joint Q values and the concatenation of independent Q values.
>
> **Ans:**
> Yes. The notation 'Err' stands for the excess risk that is the error gap between the solution we find and the ground truth optimal solution. Therefore the second term is the best achievable error on the population level and we are not sure we can find the optimal solution $(\mathbf{w}^*, \mathbf{Q}^*)$. The excess risk 'Err' reflects how close the solution we find $(\hat{\mathbf{w}}, \mathbf{Q}_t)$ to this optimal solution.
>
> > Q3(iii). How do we go from this line to line (5)? On a related note, how are we going from line (5) to (6)?
>
> **Ans:**
> Both Eq.(5) and Eq.(6) give the generalization bound of the excess risk. The inequality (5) gives an upper bound of the excess risk when each agent has its individual Q-function. In Eq.(6), we consider the case of parameter sharing, i.e. all agents share one deep Q network. Please see Appx.B.3 and Appx.B.4 for the complete proof of Eq.(5) and Eq.(6). Furthermore, we assume that $Q$ is a $L$-layer sparse ReLU network and $TQ$ is a composition of Holder smooth functions. Then we analyze the convergence rate of the excess risk as to the sample size $N$ and the training time $T$ tend to infinity.
>
> > Q3(iv). “The first term on the RHS of (5)...” I don’t see how this difference is related to the contribution-based role which is based on an agent’s Q values. How can this be related to the linear weights?
>
>
> **Ans:**
> The variance term $\text{Var}_n(\mathbf{Q}^*)$ reflects the Contribution-based Role. But the variance term is non-optimizable since $\mathbf{Q}^*$ is the optimal solution under the population level. Therefore, if the contribution-based role diversity of all agents is significant, i.e. $\text{Var}_n(\mathbf{Q}^*)$ is large, the credit assignment can reduce the excess risk by minimizing $\|\mathbf{w}^*-\hat{\mathbf{w}}\|$. Here we only consider the generalization properties and do not take the optimization procedure of $\hat w$ into consideration. Please refer to the remark at the end of Page 18 for more discussions.

---

> ### Author Response · Authors · 2021-11-19
> **Response to Reviewer L6y7 Part 2**
>
> --- method ---
>
> > Q2(i). How does this proposal show that agents take on the same role? ...The policy-based role also seems to have a lot of overlap with the trajectory-based role
>
> The definition of three different role types covers not only the factor like action frequency or observation overlap which is used to calculate the role diversity in Sec.3, but also implicitly reflects the agent’s other attributes like attack behavior, move speed, and character (brave, timid). For example, the location information can be included tractory-based role.
>
> Overlap exists in the policy-based role and the trajectory-based role. However, they can not be boiled down into one role. In Table 1 and Table 2, we find some scenarios that have relatively small policy-based role diversity but large trajectory-based role diversity, e.g. 3s_vs_5z. This indicates that these two metrics can be very different. We collected metric data of all three role diversities from different scenarios in Table 4. And discussions are held in Appx.E to study the connections between them.
>
> > Q2(ii). In Equation 1, a sum over the policy conditioned on action is an unnormalized probability distribution. It’s not clear to me how this is being used in the KL metric.
>
> **Ans:** No. It is a mean not a sum over the policy in Eq.1. Therefore, the probability distribution is normalized and can be used in the KL metric.
>
> >Q2(iii) In Equation 2, the normalization should happen over A^2, and not A.
>
> **Ans:**
> No. The normalization should happen over A. We used symmetrical KL divergence to calculate distance between agent pairs and the same agent pair should only be calculated once.
>
> > Q2(iv). It seems weird that the trajectory-based role is defined as the observation overlap. ..., and less about roles.
>
> **Ans:**
> The observation overlap is a good choice for measuring the tractory distance and is general for many MARL tasks. We give examples in Appx.C, covering both game (SMAC) and real-world tasks (multi-agent VLN) with three different methods including radius-based overlap, semantic-based overlap, and image-based observation overlap to calculate the trajectory distance.
>
> > Q2(v). I disagree with the authors on their interpretation of Q values on “contribution-based roles”: ... the authors should elaborate.
>
> **Ans:**
>
> We claim that the Q value is highly related to the contribution of the agent. It is common sense in RL that action-state pairs with a higher reward will be given a higher Q value. This still holds in MARL with an additional credit assignment module that assigns the global reward signal to individual agents to optimize the individual policy. Agents with more contributions are expected to be awarded more, which raises the Q value estimate. This is also proved by the Q value curves in Figure 18. For instance, 1c1s1z_vs_1c1s3z has a large contribution diversity in the first 20 timesteps in which agent 0 has a significantly larger Q value. This indicates that agent 0 (Colossus) contributes more to the team reward, which is in line with the fact that Colossus is a strong unit in Starcraft II with range group attack ability, causing more damage to the enemy compared to the other 2 units (stalker and zealot).  Other scenarios like 1s1m1h1M_vs_3/4/5z, also have large Q value diversity, which comes from the agents’ various contributions to the team.
>
> Besides, the individual contribution is not only based on the agent itself. As the game becomes harder (from 1s1m1h1M_vs_3z to 1s1m1h1M_vs_5z), the Q value curve of agent 0 decreases, which indicates that the contribution-based role is not fixed. Learning how to coordinate with the team is also an important factor in the contribution-based role. For instance, agents may sacrifice their own health to protect low-health allies, avoiding heavy punishment on team rewards.
> In a conclusion, the contribution-based role reflects the contribution each agent makes in the expected return, which is also highly related to the output Q value.

---

> ### Author Response · Authors · 2021-11-19
> **Response to Reviewer L6y7 Part 1**
>
> We are thankful for your time and extensive comments. But we strongly disagree with your comments over our paper and wish to clarify your several misunderstandings here. To our best understanding, some comments arise from misinterpreting our motivation and theoretical results. The point-by-point response is as follows:
>
>
> --- motivation ---
>
> > Q1(i). I’m not convinced that better understanding roles will lead to improvements in multiagent coordination...but not that this understanding results in coordination improvements.
>
> **Ans:**
> In this paper, we focus on finding the reason why the performance of different MARL training strategies varies in different scenarios. That is our key point and main contribution. We want to better understand the task difference and give proper metrics to describe this difference, and finally, find the connection between this description and the model performance theoretically and experimentally. We are not developing metrics to improve one MARL algorithm’s performance, like modifying the training pipeline or proposing a new regularization term. We are discussing all kinds of MARL algorithms combined with different training strategies(parameter sharing, communication, and credit assignment), and explaining what may bring improvement or degradation to the policy performance.
>
> > Q1(ii). What concrete challenges currently exist in multiagent optimization ... and how can these challenges be addressed by understanding how groups/teams can be factorized into roles?
>
> **Ans:**
> Sec.4 and theoretical analysis in Appx.B have analyzed the connection between the MARL optimization process and the role definition. The experiment further proves that a strong connection between model performance and role diversity exists. With the role concept to describe each scenario, we can avoid the issue of “policy degradation”. For instance, in scenarios that have large policy-based role diversity, fully sharing the parameter will bring performance degradation compared to the no sharing strategy. This conclusion can be drawn both from Eq.(6) and Sec.5.1. In Eq.(6), the first term of RHS indicates that forcing agents to share the Q function can enlarge the learning error when the optimal individual Q function is significantly different. In Sec.5.1, the experiment proves that using the improper parameter sharing strategy can lead to performance degradation.
>
> > Q1(iii). I’m confused why the authors break down MARL cooperative algorithms by “parameter sharing”, “communication” and “credit assignment”...
>
> **Ans:**
> Because these three components are important for MARL. They significantly impact the model performance and are the main topics in recent cooperative MARL research.
>
>
> > Q1(iv). The authors directly propose their three different role definitions in the second to last paragraph of the introduction ... It’s not clear what challenges they address and why these definitions are ones we should use for characterizing multiagent behavior.
>
> **Ans:**
>
> The role concept is important but the role definition is still unclear in MARL. We mentioned this in both Sec.1 and Sec.3. This leads us to create a new definition for the role concept and use it to describe the multi-agent system comprehensively. We propose three role definitions covering different agent behavior aspects in Sec.3 including policy-based, trajectory-based, and contribution-based. The policy-based role focuses on the frequency of actions. The trajectory-based role focuses on the trajectory distance between agents. The contribution-based role focus on the contribution that agents make. Different role types have inside connections to each other which are proved in Sec. 4, but they are also relatively independent that can not be boiled down to one as discussed in Appx.E.
>
> Using our role definition,  we can easily describe and categorize different MARL tasks. We can explain why the performance of different MARL training strategies varies in different scenarios. This is again our main contribution: we are not making improvements to current MARL algorithms or addressing any MARL challenge. We are establishing new metrics to explain why the performance varies on different tasks and which training strategy is suitable for the current task. And we prove that the role concept can do this from both theoretical and experimental perspectives.

---

> ### Comment · Reviewer_L6y7 · 2021-11-29
> **Updated response**
>
> I want to first thank the authors for their helpful clarifications, and also apologize for my belated response! After reading the authors’ response, I believe a lot of my confusion arose from the use of ‘role’ in their work, when (in my opinion) ‘metric’ is the better term to use, eg. Here: “The trajectory-based role focuses on the trajectory distance between agents”. This use of ‘role’ is quite different from the multiagent systems literature I’m more familiar with, which primarily refer to the question of task division/allocation. I do think the authors should take this into consideration and make this distinction explicit.
>
> I have increased my score, however, I have qualms about the paper framing/writing and would recommend against the use of ‘roles’ - it feels quite conflated in this context, and led to my confusion when reading the paper. After reading through the other reviewers’ responses, it seems like all the reviewers also equated the use of ‘roles’ as ‘metrics’. To avoid possibly confusion / conflict with previous MAS systems on roles (ie task allocation across team members), I recommend a different framing.
>
> > “We are discussing all kinds of MARL algorithms combined with different training strategies(parameter sharing, communication, and credit assignment), and explaining what may bring improvement or degradation to the policy performance.”
>
> I see, I like this formulation a lot more. After re-reading the paper, I think that actually introducing “Role Diversity” as the core term is a bit distracting. It was not immediately clear to me when reading the paper because a lot of emphasis is put on how these training strategies affect role diversity: the paper is framed to validate the connection between role diversity and the training strategies.
>
> This is related to why I found the role definitions confusing, because these are not really ‘roles’ and rather metrics, eg. the use of ‘role’ feels conflated in this sentence when metric should rather be used: “The trajectory-based role focuses on the trajectory distance between agents”.
>
> > “We claim that the Q value is highly related to the contribution of the agent. …This is also proved by the Q value curves in Figure 18. “
> If this claim is correct, I find it odd that the agent contributions/ordering more or less stay the same throughout the episode. Could the authors elaborate?

---

> > ### Author Response · Authors · 2021-11-30
> > **Response to Reviewer L6y7**
> >
> > Thank you very much for your response.
> >
> > > I have qualms about the paper framing/writing and would recommend against the use of ‘roles’... I recommend a different framing.
> >
> > Thank you for your suggestion. We are aware of the difference between our work and existing works, i.e. the definition of 'role' and the metrics of role diversity. Therefore, we emphasized our motivation at the beginning of the introduction and used a lot of space to explain how the diversity of roles affects MARL. In the revision, we will further refine the framework of the paper to avoid possible confusion.
> >
> > > This is related to why I found the role definitions confusing, because these are not really ‘roles’ and rather metrics, eg. ...”.
> >
> > In our paper, role is a concept, role diversity is a metric. For instance, trajectory-based role can not be directly used to describe a multi-agent system.  Only the role diveristy based on different role definition can do this. We will highlight this in the revision.
> >
> > > “We claim that the Q value is highly related to the contribution of the agent. …If this claim is correct, I find it odd that the agent contributions/ordering more or less stay the same throughout the episode.  Could the authors elaborate?
> >
> > Two explanations on this phenomenon:
> >
> > - contribution-based role is highly related to agent type, which is fixed through the whole game (figure.18 e,f,g,h).
> > - contribution-based role is impacted by the agent's initial attributes like position or health (figure.18 a,b,c,d)
> >
> > Therefore, in most cases, the order of contribution stays the same. We hope this may clarify your doubt.

---

### Official Review · Reviewer_97k6 · 2021-11-03

**Correctness:** 3
**Technical Novelty And Significance:** 3
**Empirical Novelty And Significance:** 3
**Recommendation:** 6
**Confidence:** 4

**Main Review:**

Strength:
+ The paper provides an interesting perspective (role-diversity) to measure the difference between MARL tasks.
+ They analyze the impact of the role diversity on MARL methods both theoretical and experimental.  The experimental results, measured by the three role diversity metrics, answered a common concern question that why the MARL model performance varies across different tasks.
+   The discovered guidelines about the choice of training strategies will benefit the community.

Weakness:
- The generalization of the proposed metrics. The trajectory-based role relies on the observation overlap percentage. The observation overlap percentage can be easily defined and measured in SMAC, but how to measure it in other realistic scenarios, e.g., embodied vision robot? The contribution-based role highly depends on the output of the RL model. However, there are a great number of hyper-parameters that will impact the output, and different methods may be of different hyper-parameters. So, how to standardize these metrics for a fair comparison?
- The experimental results are unclear. For example, in Table 1, only the policy-based role diversity is reported in MPE. Why not report the three metrics jointly for a more comprehensive analysis?  similar problems do also exist in Table. 2 and .3. These can help us better understand the difference between the three metrics.

**Summary Of The Paper:**

This paper study the relation between role diversity of tasks and the MARL model performance. The role diversity (difference among agents) is described from three perspectives: policy-based, trajectory-based, and contribution-based. The authors analyze how the role diversity impacts the MARL performance in theory and experiments. The empirical results show a strong relation among the three metrics and the three main topics in MARL (parameter sharing, communication, and credit assignment).

**Summary Of The Review:**

The paper is well-motivated and interesting. The provided guidelines are practical. But I am concerned about the generalization of the proposed metrics in other MARL environments.

---

> ### Author Response · Authors · 2021-11-19
> **Response to Reviewer 97k6 Part 2**
>
> > Q2. The experimental results are unclear. For example, in Table 1, only the policy-based role diversity is reported in MPE. Why not report the three metrics jointly for a more comprehensive analysis? Similar problems do also exist in Table. 2 and 3. These can help us better understand the difference between the three metrics.
>
> **Ans:**
>
> Thank you for the suggestion. We will discuss this from both theoretical and experimental perspectives.
>
> From theoretical analysis, there does exist some relation between different role diversity. The contribution-based role diversity is a compound description of role diversity. It corresponds to the variance term in equation (5). For the parameter sharing case, we decompose this variance into a sum of two terms: a bias term corresponds to the policy-based role diversity and a variance term corresponds to the trajectory-based role diversity. Therefore, under the simple scenario in Sec. 4, we can find a clear relationship between different role diversity.  But for more complicated settings, the decomposition in (6) may not hold. We have discussed this issue in the remark on Page 18.
>
> From the experiment perspective, we collected all related data in Table 4 and have further discussion in Appx. E. In conclusion, in our experiment, different role diversity metrics are relatively independent of each other.  This may be because real MARL tasks or experiment platforms are more complicated and these relations may not be very significant as theoretical analysis. Still, strong relations to different aspects of MARL training strategies exist, including parameter sharing, communication, and credit assignment, which is the main contribution of our paper.

---

> > ### Comment · Reviewer_97k6 · 2021-12-01
> > **Response to Authors**
> >
> > First, I apologize for my belated response. I thank the authors for their detailed response. Some of my concerns are addressed in the response, e.g., the fairness of the contribution-based metric. However, I am still concerned about the generalization of the observation-based metric in other scenarios. Particularly, after reading Appendix C.2 and C.3, I think it is not a good idea to directly compute the similarity of different views in feature space for embodied agents. Instead, some 3D vision methods (3D reconstruction/ SLAM) should be considered for a more general and fair measurement. So I prefer to keep my original score, as the paper is interesting but still has room to improve.

---

> > > ### Author Response · Authors · 2021-12-01
> > > **Response to Reviewer 97k6**
> > >
> > > Thank you very much for your response.
> > >
> > > In this paper, role and role diversity are defined for the analysis of the connection between role diversity and MARL optimization. Therefore, we want these definitions to be neat, extendable, and easy to calculate.
> > >
> > > However, we are not focusing on building a perfect role definition that can match every different multi-agent scenario. It is so challenging because the task settings of different MAS vary a lot. Imagine how different the role diversity is in the matrix world compared to 3D embodied robot navigation tasks. We are sorry that we can not discuss how to compute trajectory-based role diversity with 3D reconstruction/SLAM techniques in this paper as it is out of the scope of MARL，and is not related to the key point of this paper.
> > >
> > > We want to emphasize the main contribution of this paper: to study the relationship between policy performance and task characteristics， which can be described by the role diversity. The role diversity should be measured from different aspects to ensure the accuracy and completeness of the description.
> > >
> > > Thanks for your suggestions and any further concern is welcome.

---

> ### Author Response · Authors · 2021-11-19
> **Response to Reviewer 97k6 Part 1**
>
> We are thankful for your time and insightful comments. We are particularly grateful for your recommendations to improve the manuscript. We answer your questions as follows.
>
> > Q1(i). The generalization of the proposed metrics. The trajectory-based role relies on the observation overlap percentage. The observation overlap percentage can be easily defined and measured in SMAC, but how to measure it in other realistic scenarios, e.g., embodied vision robot?
>
> **Ans:**
>
> The reviewer points out that trajectory-based role diversity can be easily calculated in game-based environments like SMAC using observation overlap as defined in Sec. 3.2, but this measurement cannot be applied to other multi-agent scenarios like embodied vision robot tasks.
>
> The advantage of using observation overlap to measure the trajectory distance is obvious: the raw trajectories may have different lengths (agents are expected to stop or start at different time steps) and the distance between them is hard to normalize to a fixed range (while the observation overlap has a fixed range from 0 to 1). Still, we agree with the reviewer that the generalization of this metric should be further discussed. Here we claim that the ‘overlap’ concept is general from both theoretical and applicable perspectives and is extendable to other tasks.
>
> From the theoretical perspective, we provide a clear decomposition for the MARL learning object in Sec.4 based on a simple and common setting in MARL tasks that each agent makes individual observations and the learning procedure of all agents is also independent. The impact of trajectory diversity (based on observation overlap) can be observed from this decomposition. In Eq. (6), the variance $\text{Var}_n\big(\bar{\mathbf{Q}}^*(\mathbf{z}, \mathbf{u})\big)$ only comes from the different observations and actions of agents. If all agents share the same observation by the communication mechanism, the variance term can be effectively reduced. However, sharing observation via communication slows down the convergence rate which is proved in Appx.B.5. These form a dilemma on how we balance the variance term and learning speed, which is related to trajectory-based role diversity.
>
> From the applicable perspective, we extend our metric to a more complicated real-world task: multi-agent vision language navigation (VLN) to show that the ‘overlap’ concept is extendable to other tasks. The VLN task takes real 2D images and language descriptions as input. Please find the detailed illustration in our paper Appx.C.2.and C.3.  Although the settings of VLN tasks are significantly different from SMAC and MPE tasks, we can still manage to utilize the ‘overlap’ concept for calculating the trajectory distance and measuring the role diversity inside the task.
>
> In conclusion, different multi-agent systems have different ways to compute d_t in Eq.(3) based on our proposed ‘overlap’ concept. We are not claiming that observation overlap used in SMAC  is the best way to measure the trajectory-based role diversity but to emphasize that it could be a wise choice compared to directly computing the distance between different raw trajectories, which is troublesome in most cases.
>
> > Q1.(ii) The contribution-based role highly depends on the output of the RL model. However, there are a great number of hyper-parameters that will impact the output, and different methods may be of different hyper-parameters. So, how to standardize these metrics for a fair comparison?
>
> **Ans:**
>
> This is also a great point. Although Table 3 and Figure 16 show a stable range of contribution-based role diversity from 0.5 to 2.0, we are still in one typical domain (SMAC), where the output of the RL model is relatively stable. As the reviewer said, changing the hyper-parameters or moving to another MARL environment can have a direct impact on the scale of this metric.
> To overcome this, we modify Eq.4 with a max function to keep the contribution-based role diversity scale in 0 to 1, which is similar to trajectory-based role diversity. In this way, the value of contribution-based role diversity becomes less sensitive to hyper-parameters, reward function, and environmental change. Related experiment results are re-provided in Table 3 on the Q diversity column.
>
> And also, no matter the scale is fixed or not, the role diversity is not a fixed value in most cases. For instance, in Figure 2 (b)(c), the role diversity curves float in a small range. This comes from the random seed in the training stage and the inherent randomness of the environment (like the enemy attacking order in SMAC). However, it can still provide us a good description of MARL tasks as shown in Sec. 5.

---

### Author Response · Authors · 2021-11-19
**General Response**

Dear reviewers,

Thank you for your careful analyses of our work and valuable feedback. We have uploaded a new version of the paper incorporating your feedback and numerous quality improvements. The main changes are summarized below.


- We revise Sec. 4 with a summary at the end of this Sec. to emphasize the relation of role diversity to the MARL learning process from the theoretical perspective.
- We add Appx. C.2 and C.3 to illustrate the generalization of the trajectory-based role diversity.
- We revise Appx. E to discuss the connection between different role types from the experimental perspective.
- We revise Appx. F, introducing SEPS to MARL knowledge sharing. SEPS uses a selective parameter sharing strategy across agents, which serves as a complement to the ‘full/partly/separate’ methods in our paper. Further discussion and experiment results can be found in F.1
- We normalize the contribution-based role diversity to (0,1) in Eq.4
- We correct typo errors and revise confusing terms in our paper.

Also, we would like to address the major concern raised by all four reviewers here, which is focusing on the necessity and validity of our role definition.

Although the role concept has been used in natural systems [1][2], how to utilize it in MARL is still under investigation. Recent works [3][4][5][6] find that using emergent role groups or a predefined role structure can facilitate policy learning. [3][4] define the role concept based on the agent’s trajectory. [5] uses action embedding with unsupervised clustering to get the number of role groups. [6] proposes a VAE based method that automatically groups the agent based on the action impact on the environment with a random policy. Methods are diverse but are all aimed at grouping the agents for better coordination.

The motivation of our work is significantly different from existing works.

First, we are not developing a new MARL algorithm based on the role definition to surpass other algorithms. On the contrary, we claim that there is no single algorithm that fits all tasks. The question is how we choose the best one for the current task. There are two steps in our solution: 1. developing a metric to describe the task and 2. according to the task description, choosing the proper training strategies.

Second, it is still unclear how much information needs to be included from the multi-agent system to the role definition before. Barely using the trajectory information or the action embedding is not enough like [3][4][5]. For instance, none of them can clearly describe the agent type or the team contribution percentage. To comprehensively describe the role concept, we, therefore, establish the policy-based, trajectory-based, and contribution-based role type, along with diversity calculation methods for each role type. Different role diversities have different impacts on parameter sharing, communication, and credit assignment strategies choosing. We proved it from both theoretical and experimental perspectives in our paper. Finally, based on the role diversity description, we are able to choose the proper training strategy combination for the current task.

[1] Emergence of Division of Labour in Halictine Bees: Contributions of Social Interactions and Behavioural Variance

[2] The Organization of Work in Social Insect Colonies

[3] ROMA: Multi-Agent Reinforcement Learning with Emergent Roles

[4] Coordinated Multi-Agent Imitation Learning

[5] RODE: Learning Roles to Decompose Multi-Agent Tasks

[6] Scaling Multi-Agent Reinforcement Learning with Selective Parameter Sharing

---

### Decision · Program_Chairs · 2022-01-20

**Decision:**

Reject

**Comment:**

This paper proposes a Role Diversity metric, meant to quantify how different roles are in a multi-agent RL setting. There's actually three versions of this metric, or three aspects (the distinction is not entirely clear to this area chair).

The reviewers are generally not very enthusiastic about the paper, with scores hovering at or just below the acceptance threshold. There has been extensive discussion between reviewers and authors, but there a sense that there is confusion about the exact purpose and contribution of the paper. This is reinforced by the authors' "letter to area chair", which outlines several ways the reviewers have not gotten the message. Reading the paper, it appears to me that the root cause is that the authors are indeed not communicating clearly what the paper contributes and why. It is, after all, the authors' responsibility that the reviewers understand the work. My own impression is that the text is dense and not particularly easy to get through. Perhaps the authors are simply trying to cram too many contributions into a single conference paper? This is a classic error which leads to hard-to-read papers. In addition to this, there is a lingering concern about the generalizability of the proposed methods.

I think the authors need to work more on their presentation, and perhaps reconsider which parts to include in their paper and exactly which measure they want to send, before they submit to another venue.